

# Emissions Preparation and Analysis for Multiscale Air Quality Modelling over the Athabasca Oil Sands Region of Alberta, Canada

Junhua Zhang[1], Michael D. Moran[1], Qiong Zheng[1], Paul A. Makar[1], Pegah Baratzadeh[2],
George Marson[3], Peter Liu[1], and Shao-Meng Li[1]

[1]Air Quality Research Division, Environment and Climate Change Canada, 4905 Dufferin Street, Toronto, ON, M3H 5T4, Canada
[2]Pollutant Inventories and Reporting Division, Environment and Climate Change Canada, 4905 Dufferin Street, Toronto, ON, M3H 5T4, Canada
[3]Air Quality Research Division, Environment and Climate Change Canada, 335 River Road, Ottawa, ON, K1A 0H3, Canada

*Correspondence to:* junhua.zhang@canada.ca

**Abstract.** The oil sands of Alberta, Canada are classified as unconventional oil, but they are also the third-largest oil reserves in the world, behind only Venezuela and Saudi Arabia. We describe here a six-year effort to
improve the emissions data used for air quality (AQ) modelling of the roughly 100 km x 100 km oil extraction and processing industrial complex operating in the Athabasca Oil Sands Region (AOSR) of north-eastern Alberta. The objective of this work was to review the available emissions data, provide information for comparison with observation-based emissions estimates, and generate model-ready emissions files for the Global Environmental Multiscale–Modelling Air-quality and CHemistry (GEM-MACH) AQ modelling system
for application to the AOSR. GEM-MACH was used to produce nested AQ forecasts during an AQ field study carried out in the AOSR in summer 2013 as well as ongoing experimental forecasts since then and retrospective model simulations and analyses for the field-study period. This paper discusses the generation of GEM-MACH emissions input files, in particular for a high-resolution model domain with 2.5-km grid spacing covering much of western Canada and centred over the AOSR. Prior to the field study, ten pre-2013 national, provincial, or
sub-provincial emissions inventories for up to seven criteria-air-contaminant species ($NO_x$, VOC, $SO_2$, $NH_3$, CO, $PM_{2.5}$, and $PM_{10}$) that covered the AOSR study area and that had been compiled for various purposes were reviewed, and then a detailed hybrid emissions inventory was created by combining the best available emissions data from some of these ten inventories. After the field study, additional sources of emissions-related data became available, including 2013 hourly $SO_2$ and $NO_X$ emissions and stack characteristics for large point
sources measured by Continuous Emission Monitoring Systems, 2013-specific national inventories, daily



reports of SO$_2$ emissions from one AOSR facility for a one-week period during the field campaign when that facility experienced upset conditions,  aircraft measurements of VOC and PM2.5 concentrations from the 2013 field campaign and derived estimates of their emissions, and measurements of chemical composition of dust collected from various AOSR sites.  These new data were used to generate updated emissions input files for

various post-campaign GEM-MACH sensitivity studies.  Their inclusion resulted in some significant emissions revisions, including a reduction in total VOC and SO2 emissions from surface mining facilities of about 40% and 20%, respectively, and a ten-fold increase in PM$_{2.5}$ emissions based on aircraft observations.  In addition, standard emissions processing approaches could not provide an accurate representation of emissions from such large, unconventional emissions sources as AOSR surface mines.  In order to generate more accurate high-

resolution, model-ready emissions files, AOSR-specific improvements were made to the emissions processing methodology.  To account for the urban-scale spatial extent of the AOSR mining facilities and the high-resolution 2.5-km model grid, novel facility-specific gridded spatial surrogate fields were generated using spatial information from GIS (geographic information system) shapefiles and satellite images to allocate emissions spatially within each mining facility.  Facility- and process-specific temporal profiles and VOC

speciation profiles were also developed.  The pre-2013 vegetation and land-use data bases normally used to estimate biogenic emissions and meteorological surface properties were modified to account for the rapid change of land use in the study area due to marked, year-by-year changes in surface mining activities, including the 2013 opening of a new mine.  Lastly, mercury emissions data were also processed to support AOSR mercury modelling activities.  The combination of emissions inventory updates and methodological improvements to

emissions processing has resulted in a more representative and more accurate set of emissions input files to support AQ modelling to predict the ecosystem impacts of AOSR air pollutant emissions.  Seven other papers in this special issue used some of these new sets of emissions input files.

## 1 Introduction

Alberta's oil sands (OS: see Table S1 for a list of acronyms), which consist of a mixture of bitumen, sand, clay,

and water, are found in the Athabasca, Cold Lake, and Peace River areas of northern Alberta.  Together these areas cover 142,200 km$^2$, about 21% of the area of the province of Alberta (Alberta Energy, 2017) or about the same area as Greece.  The Athabasca Oil Sands Region (AOSR) contributes the largest share of OS bitumen production: 82% in 2015 (Alberta Energy Regulator, 2017a).  There are two main methods used to produce oil from the bitumen, each of which has atmospheric emissions.  For bitumen deposits buried less than 200 feet



below the surface, the oil sands are mined by open-pit mining methods, in which large excavators dig up oil sand ore and transfer it to heavy-hauler trucks for transport to crushers, where large ore lumps are broken up. The crushed ore is then mixed with hot water and transported to an extraction plant, where the bitumen is separated from the other components and then transferred to either an on-site or a remote upgrader to create

synthetic crude oil. About 3% of the OS area, mainly within the AOSR, can be surface-mined but it accounts for about 20 percent of the recoverable OS oil reserves. Oil sands in the remaining 97% of the OS area are situated too deep for surface mining and can only be recovered by *in situ* extraction methods such as steam-assisted gravity drainage (Alberta Energy Regulator, 2017b). As of 2015, about 46% of Alberta oil production from oil sands comes from surface mines in the AOSR (Alberta Energy Regulator, 2017a).

According to the 2013 National Pollutant Release Inventory (NPRI; Canada's legislated inventory of pollutant releases reported by industrial, commercial, and institutional facilities that meet certain reporting requirements, emissions from Alberta's OS sector account for 61%, 34%, and 14% of the total reported VOC (volatile organic compound), $SO_2$, and NOx emissions, respectively, for the province, whose NPRI total VOC, $SO_2$, and $NO_x$

provincial emissions are the highest of the Canadian provinces (https://www.canada.ca/en/environment-climate-change/services/national-pollutant-release-inventory.html). The OS industrial sector is also a significant source of PM (particulate matter) and CO emissions. Due to the complex nature of open-pit mining and the OS oil extraction processes, pollutants are mainly emitted from the following processes: (1) exhaust emissions from off-road vehicles used for removal of the surface overburden and for excavation and transportation of the OS ore

to an extraction plant; (2) pollutants emitted from processing taking place at the extraction and upgrading plants; (3) fugitive VOC emissions from mine faces, tailings ponds, and extraction plants; (4) fugitive dust from surface disturbances such as the passage of the large vehicles belonging to the off-road mine fleets; and (5) wind-blown dust from open surfaces such as mine faces and tailings-pond "beaches". The emissions of criteria-air-contaminant (CAC) pollutants ($NO_x$, VOC, $SO_2$, $NH_3$, CO, $PM_{2.5}$, and $PM_{10}$) from *in situ* OS extraction

activities are currently believed to be lower than those of open-pit mines based on the emissions reported to NPRI by facilities (https://www.canada.ca/en/environment-climate-change/services/national-pollutant-release-inventory.html).

To support air quality (AQ) modelling activities that are part of the Governments of Canada and Alberta Joint

Oil Sands Monitoring (JOSM) Plan (see JOSM, 2011), emissions input files were created over the past six years for Environment and Climate Change Canada's (ECCC) Global Environmental Multiscale – Modelling Air-



quality and CHemistry (GEM-MACH) AQ modelling system, which was set up to conduct nested AQ forecasts at model horizontal grid spacings of 10 km and 2.5 km (see Figure S1). The generation of emission input files was particularly challenging for the inner 2.5-km grid because the AOSR surface mining and processing facilities at the centre of the grid are large, complex, and unconventional industrial facilities that cannot be well

represented by standard emissions processing approaches. At the beginning of emissions-related work for the JOSM plan in 2012 (referred to as Phase 1), considerable effort was invested in three areas: (i) the review of different available emissions inventories covering the AOSR compiled for different agencies with various geographic and temporal coverages (Alberta Environment and Sustainable Resource Development (AESRD [now AEP (Alberta Environment and Parks)], 2013; Marson, 2013); (ii) the compilation and synthesis of best-

available emissions data into a hybrid JOSM emissions inventory (ECCC & AEP, 2016); and (iii) the preparation of GEM-MACH emissions input files for multiple model grids to support AQ forecasting for an Aug.–Sept. 2013 AQ field campaign in the AOSR (Gordon et al., 2015; Liggio et al., 2016; Li et al., 2017). Particular attention was paid to the emissions input files for the inner (2.5-km) model domain centred over the AOSR, which comprised the largest share of OS bitumen production during the 2013 field study period.

Additional emissions input files were then developed for JOSM Plan AQ modelling activities in two more phases as 2013-specific emissions data became available. The three phases to support GEM-MACH development, testing, evaluation, and application can thus be summarized as follows: (1) in the first phase (2012-13), emissions input files based on older inventories were created prior to and in support of AQ forecasting for the 2013 field study and post-study analysis shortly thereafter; (2) in the second phase (2014-15),

updated sets of emissions files were created for post-study analyses based on new emissions-related information available after the field study; and (3) in the third phase (2016-17), more sets of emissions input files were created based on updated emissions inventories, as well as new emissions estimates from analysis of the 2013 field-study measurements.

GEM-MACH emissions input files developed during the first two phases have been discussed in Zhang et al. (2015) and in a joint report by ECCC and AEP (formerly AESRD) for the JOSM project (ECCC & AEP, 2016: hereinafter referred to as the JOSM report). This paper briefly summarizes the work of the first two phases but focuses on the development of new emissions input files during the third phase for the following GEM-MACH AQ modelling applications:



1) Base-case study for AQ forecasting and a long-term deposition study for the region (Makar et al., 2017, this issue) and for improvements for $NH_3$ predictions (Whaley et al., 2017, this issue);

2) Model sensitivity study on the use of CEMS (Continuous Emission Monitoring System) measurements of $SO_2$, $NO_x$, exit temperature, and flow rate (Akingunola et al., 2017b and Gordon et al., 2017 this issue);

3) Model sensitivity study on impact of updated VOC and $PM_{2.5}$ emissions and speciation derived from surface measurements and from airborne measurements made during the 2013 field campaign (Stroud et al., 2017, this issue);

4) Sensitivity study on the impact of increased model horizontal resolution down to 1 km to model predictions (Russell et al., 2017, this issue)

5) Mercury modelling over North America and the OS area using updated emissions (Fraser et al., 2017, this issue).

In the rest of this paper, Section 2 provides an overview of the comprehensive national, provincial, and subprovincial emissions inventories considered to build the base-case model emissions for all three phases. Challenges faced and approaches taken to compile a best-available hybrid emissions inventory for the three phases are discussed. Section 3 describes the emissions processing methodology applied in Phase 3 (2016-17) to generate base-case hourly, gridded emissions input files, including both anthropogenic and biogenic emissions. Unique facility-specific and process-specific spatial surrogate fields were created for six AOSR surface mines to allocate emission from these large and unconventional sources. A land-cover database was also updated for biogenic emissions and for land-surface characterization to account for the rapid change of land use over this region. Next, Section 4 describes the emissions data and emissions processing used for several emissions sensitivity studies, including those based on new hourly CEMS measurement data, on aircraft- and surface-observation-based estimates of VOC and $PM_{2.5}$ emissions and chemical speciation, and on updated mercury emissions. Lastly, Section 5 provides a summary of this work and presents plans for future updates and improvements of emissions for AOSR AQ modelling.



## 2 Emissions Inventories Used for the Base-Case Emissions

### 2.1 Review of emissions inventories used for JOSM Phases 1 and 2 AQ modelling

In 2012, prior to the summer 2013 AOSR field study (Gordon et al., 2015; Liggio et al., 2016; Li et al., 2017), the national emissions inventories used to generate the emissions input files for ECCC's operational GEM-
MACH AQ forecast model consisted of the AQ modelling version of the 2006 Canadian national Air Pollutant Emission Inventory (APEI) from ECCC, a projected 2012 U.S. National Emissions Inventory (NEI) from the U.S. Environmental Protection Agency (EPA) based on version 4 of the 2005 U.S. NEI, and the 1999 Mexican inventory (Moran et al., 2013a, 2014). The 2006 Canadian APEI represented a base year seven years earlier than the field study period, an important consideration for the AOSR due to its rapid development. For
example, one of the five AOSR surface mining facilities in operation in 2012, the Canadian Natural Resources Limited (CNRL) Horizon mine (see Figure 1), only began production in 2009. Hence, pollutant emissions from that mine were not available in the 2006 APEI. Thus, while the 2006 APEI was being used as the basis for national-scale operational AQ forecasting for Canada, it was not an ideal choice for high-resolution AQ modelling for the AOSR field study.

A number of newer emissions inventories, however, had been developed for the AOSR area or for the province of Alberta, albeit not always for the purpose of supporting AQ modelling. Prior to the 2013 AOSR field study, the ten inventories listed in Table S2 by name, target region, and base year were reviewed to choose the most suitable emissions inventory data for AQ modelling for the OS area (AESRD, 2013; Marson, 2013). After an
intensive review of these newer inventories, it became clear that no one inventory was the "best" choice in all respects, but three inventories contained emissions data that were either unique (i.e., not reported elsewhere), more detailed, and/or the most recent. The 2009/10 Cumulative Environmental Management Association (CEMA) inventory (Davies et al., 2012) had the most detailed stack- and process-level emissions for the AOSR surface mining facilities shown in Figure 1 (including separate emissions from mine faces, tailings ponds, and
off-road mine hauler fleets, except for fugitive dust emissions from the off-road fleet); the 2010 Canadian NPRI included emissions for some species ($NH_3$ and $PM_{10}$) and source types (fugitive dust emissions from OS mine fleets) missing from the CEMA inventory; and the 2010 Canadian APEI from ECCC was the most comprehensive and had the largest spatial coverage (national) for area sources. Note, however, that at this time the 2010 Canadian APEI was not yet available in the detailed format required for emissions processing (referred
to as the AQ modelling version).



### 2.1.1 Phase 1 hybrid emissions inventory and ancillary files

The solution adopted in the first phase for the best base-case inventory to use to generate GEM-MACH emissions input files for the 2013 field study was to create a synthesized or hybrid AQ modelling emissions inventory, as summarized in Table 1, that combined the best available information from the above inventories.

The 2009/10 CEMA inventory, supplemented by the 2010 NPRI for $NH_3$ and $PM_{10}$ emissions, was mainly used to provide emissions for the AOSR field-study area while the 2006 APEI was used outside the AOSR where the CEMA inventory's coverage ended. The 2010 NPRI was also used to scale the CEMA facility-total VOC emissions for the five AOSR surface mines active at that time (Figure 1), since it was found that the CEMA inventory had the lowest total VOC emissions for these five facilities compared to four other inventories (ECCC

& AEP, 2016) and the NPRI is Canada's legislated inventory of large point sources based on emissions reported by facilities. The ratio of 2010 NPRI total VOC emissions for the five mining facilities vs. the CEMA total yielded a scaling factor of 2.6, which was applied to the CEMA facility-total VOC emissions for the individual facilities (Table 2). One reason to focus on the VOC emissions from these five facilities was that for 2010 they were estimated by NPRI to have contributed 75% of VOC emissions from all Alberta facilities. The 2009/10

CEMA inventory was also used to specify the process-specific allocation of these facility-total emissions between mine faces, tailing ponds, plants, and smoke stacks, which then dictated the spatial and temporal allocation and chemical speciation of these process-level emissions (ECCC & AEP, 2016).

The focus of the OS field study was a roughly 100 km by 100 km subregion of the AOSR located north of Fort

McMurray, Alberta (Figure 1). This study area contains a complex of six large surface bitumen mining and processing facilities situated on both sides of the Athabasca River. As shown in Figure 1, each mining facility covers a very large area, ranging from 66 to 275 km$^2$, and each facility contains various area sources within their boundaries, including NOx, CO, VOC, and $PM_{2.5}$ emissions from each mine's off-road heavy-hauler fleet, evaporative VOC emissions from tailings ponds and mine faces, and point sources of $SO_2$, NOx, CO, VOC,

$PM_{2.5}$ and fugitive VOC emissions from extraction and upgrading plants (Zhang et al., 2015). Although emissions from industrial facilities are normally treated as point sources by emissions processing systems and AQ models (e.g., Houyoux et al., 2000), each of these six facilities spans more than 10 GEM-MACH 2.5-km grid cells (area of 6.25 km$^2$ each), and many of the emissions are distributed over large areas within the facility boundaries. Treating such large facilities as point sources that can be assigned to a single grid cell is thus not

realistic.



To address this concern, a new approach was taken in which these nominal point sources were treated as area sources. First, a GIS shapefile based on data collected by AESRD was obtained for the year 2010 with detailed locations of mine faces, extraction plants (and, for three facilities, upgrading plants), and tailings ponds for the five AOSR mines that were active within the study area at that time: Suncor Millenium and Steepbank mines;

Syncrude Mildred Lake mine; Syncrude Aurora North mine; Shell Canada Muskeg River mine and Jackpine mine (known collectively as the Shell Canada Albian Sands mine); and CNRL Horizon mine (Figure 1). This shapefile was then used to develop three spatial surrogates for each facility to be used for spatial allocation of mine-face, tailings-pond, and extraction/upgrading plant emissions, respectively, including emissions from the off-road mining fleet and evaporative VOC emissions from mine faces, extraction plants, and tailing ponds

(Zhang et al., 2015). It was assumed that the off-road fleets operated mainly in the mine-face areas, so the mine-face spatial surrogate field was used to allocate CAC emissions from the off-road fleet as well as evaporative VOC emissions from the mine faces. Note that emissions from the main smokestacks of the facilities were still treated as point sources. Finally, once all of the above development work was completed, the hybrid Phase 1 emissions inventory was input to the SMOKE (Sparse Matrix Operator Kernel Emissions)

emissions processing system (https://www.cmascenter.org/smoke) together with the new AOSR facility-specific spatial surrogate fields to generate Phase 1 model-ready emissions input files for use by GEM-MACH during the 2013 summer field study (Zhang et al., 2015).

**2.1.2 Phase 2 hybrid emissions inventory and ancillary files**

In Phase 2, after the field study, emissions updates were made during the 2014-2015 period to include newly

available emissions information, including (i) an AQ modelling version of the 2010 Canadian APEI, (ii) a preliminary version of the 2013 NPRI point-source inventory, (iii) stack-level continuous emission monitoring system (CEMS) measurements for 17 smokestacks at four AOSR mining facilities for the field-study months of August and September 2013, and (iv) daily reports of $SO_2$ emissions during abnormal operating conditions from one AOSR mining facility (CNRL Horizon) during a one-week period in August 2013 when up to 20 times

normal daily $SO_2$ emissions were released to the air during several upset events (ECCC & AEP, 2016). The six inventories and other emissions data sources that were used to create a second hybrid Canadian AQ modelling emissions inventory for 2013 are listed in Table 3.

The GIS shapefile describing the OS mines was also updated using 2013 satellite imagery (Zhang et al., 2015).

These shapefile updates captured growth in the boundaries of existing mine faces and tailings ponds as well as



new mine faces and tailings ponds that had been opened post-2010, and they were used to update the facility-specific spatial surrogate fields. In addition, a sixth mine, the Imperial Oil Kearl mine, entered production in 2013 (see Figure 1). Annual emissions estimates for this facility were obtained from the preliminary 2013 NPRI and three new spatial surrogates were developed to allocate emissions from this facility (Zhang et al., 2015). As

well, monthly facility-specific bitumen production data reported to the province of Alberta for 2013 for the six OS mining facilities were used to create facility-specific monthly temporal profiles (Alberta Energy Regulator (AER), 2014; Zhang et al., 2015). Note that a more comprehensive and detailed description of the Phase 2 hybrid inventory, the updated ancillary data sets for emissions processing, and the emissions processing procedure that was followed with the SMOKE system to generate model-ready emissions input files using the

Phase 2 inventory is available in the JOSM report (ECCC & AEP, 2016).

## 2.2 Inventory updates for Phase 3 hybrid emissions inventory

After the generation of the Phase 2 emissions input files for GEM-MACH, five important new sources of 2013-related emissions data became available:

1)   2011 U.S. NEI Version 1 from U.S. EPA (Eyth et al., 2013);

2)   a) 2013 Canadian APEI Version 1 from ECCC for all sectors, including the first version of reviewed, publicly-available 2013 NPRI (released December 2014), except for on-road and off-road mobile source emissions (Sassi et al., 2016);

      b) Second version of reviewed, publicly-available 2013 NPRI (released December 2015)

3)   2011 Canadian upstream oil and gas (UOG) point-source inventory for small and medium UOG facilities

(Clearstone Engineering Ltd., 2014a,b,c) and a projected 2013 Canadian UOG inventory (created by ECCC as part of the 2013 APEI Version 1);

4)   CEMS measurements for all CEMS stacks with relatively large $SO_2/NO_X$ emissions in the province of Alberta for August and September, 2013 (from AEP);

5)   Aircraft-measurement-based estimates of VOC emissions during the 2013 field study period for four of the

six AOSR mining facilities (Li et al., 2017) and aircraft-measurement-based size-resolved PM emissions for all six facilities.

There were large differences noted between the 2011 U.S. NEI and the older projected 2012 U.S. NEI (projected from the 2005 U.S. NEI), despite the one-year difference in base year. For example, the projected 2012 NEI

$SO_2$ emissions from all sectors were reduced by 48% in the 2011 NEI, but $NO_2$ emissions increased in the latter





by 8%, due mainly to a 40% increase of on-road NOx emissions (Moran et al., 2015). Among the many reasons that may have contributed to the large differences between the two inventories, one is the change in on-road emissions estimation tool used by the U.S. EPA from MOBILE6.2+MOVES2010 (U.S. EPA, 2010) to SMOKE-MOVES2014 (U.S. EPA, 2015; Choi, 2016). Given that the 2011 U.S. NEI is a retrospective

inventory based on actual activity data and CEMS data for base-year 2011, it was chosen to replace the projected 2012 U.S. NEI used in Phases 1 and 2 for the creation of the Phase 3 emissions input files.

The first AQ modelling version (i.e., SMOKE-ready version) of the 2013 Canadian APEI (v1), which included point-source emissions from the first version (v1) of the reviewed, publicly-available 2013 NPRI (released in

late 2014), became available in early 2016 for most sectors, with the exception of the on-road and off-road mobile source sectors. There are significant differences for some sectors between the modified 2010 APEI used in Phase 2 (Table 3) and the 2013 APEI. Figure 2 shows a comparison of fugitive-dust $PM_{2.5}$ emissions from four sectors for the province of Alberta. $PM_{2.5}$ emissions from construction more than doubled from 2010 to 2013 due to a combination of increased construction activities and changes in the methodology used to estimate

PM emissions for this sector (Environment Canada, 2014). Table 2 provides a comparison of facility-total VOC emissions for the six surface OS mining facilities used for Phases 1/2 vs. Phase 3. For Phases 1 and 2 these emissions were 2010-NPRI-scaled CEMA VOC emissions (Tables 1 and 3), whereas for Phase 3, version 2 (v2) of the 2013 NPRI, which became available in late 2015, was used (Table 4). VOC emissions from the Suncor Millenium/Steepbank facility were reduced from about 28,000 tons/year in Phase 2 to 9,500 tons/year in Phase

3, a 64% reduction; the Shell Canada Muskeg River/Jackpine mine had a similar percentage reduction. One additional complication is that facilities may submit modified reports to NPRI for past reporting years based on updated information, as can be seen by comparing the last two columns of Table 2, where reported total VOC emissions increased for Suncor Millenium/Steepbank, Syncrude Mildred Lake, and Syncrude Aurora North in the 2013 NPRI v2 (see also Li et al., 2017). One other important change evident in Table 2 is the inclusion of

emissions from the Imperial Oil Kearl surface mine, which began production in 2013, in the two 2013 emission inventory versions.

Emissions from smokestacks that are released at high volume flow rates and temperatures may rise much higher into the atmosphere than stack releases with lower volume flow rates and temperatures. As a consequence, AQ

models such as GEM-MACH include specialized parameterizations to calculate this plume rise (see Akingunola et al., 2017b; Gordon et al., 2017, this special issue). However, the extent to which this information is reported



depends on the regulatory environment. One limitation of the 2013 NPRI is that only emissions from stacks higher than 50 m must be reported separately. Emissions from all other shorter stacks are aggregated together with surface-level fugitive emissions and are treated as surface releases (ECCC, 2016). On the other hand, the 2009/10 CEMA inventory has separate emissions information for all individual stacks. To allow plume rise to

be calculated for stacks both above and below the NPRI reporting threshold, facility-total NPRI aggregate stack emissions were allocated proportionately to each stack in the CEMA inventory based on the 2009/10 CEMA stack emissions.

There are a variety of activities with pollutant releases to air within any given facility's boundaries, and the type

of activity may influence the type and amount of VOCs being emitted at the facility. The extent to which these activities can be identified to allow spatial allocation within a facility once again depends on the regulatory environment and the reporting requirements. Surface-level fugitive VOC emissions are reported to NPRI as facility-*total* emissions without differentiation between source type (i.e., mine faces, tailings ponds, and extraction/upgrading plants). To distribute 2013 NPRI fugitive VOC emissions spatially within an OS mining

facility, process allocation factors calculated from the process-specific fugitive VOC emissions in the 2009/10 CEMA inventory for each AOSR mining facility were used to allocate fugitive VOC emissions between mine faces, tailings ponds, and plants (similar to the procedure used in Phase 2; see ECCC & AEP, 2016). For the Imperial Oil Kearl mine, which was not operating in 2010, 2013 fugitive VOC emissions were differentiated based on process allocation factors from the Shell Muskeg River/Jackpine facility given that both facilities use

Paraffinic Froth Treatment (PFT) technology to produce diluted bitumen, which is then transported through pipelines to off-site refineries for further processing (http://www.oilsandsmagazine.com/technical/mining/froth-treatment/paraffinic; Li et al., 2017).

The UOG emissions input files generated for Phase 2 were based in part on a year-2000 Canadian UOG

inventory projected to 2010 (Table 3). After Phase 2, a 2011 Canadian UOG inventory that was compiled for ECCC became available (Clearstone Engineering Ltd., 2014a, b, c). This new subinventory was then projected by ECCC to 2013 for inclusion in the 2013 APEI. Figure 3 shows the national-level differences between the year-2000-based 2010 UOG inventory and the year-2011-based 2013 UOG inventory for the seven CAC pollutants, where about 95% of the UOG facilities are located within the high-resolution OS modelling domain.

VOC, CO, and NOx emissions are higher for the new subinventory by 27%, 23%, and 11%, respectively, while $SO_2$ emissions are 11% lower. Thus, the projection of total UOG emissions from 2000 to 2010 that was used



for Phase 2 seems to have been reasonable in total. However, the number of UOG facilities with CAC emissions increased from about 207,000 in the 2000 UOG inventory to 334,000 in the 2011 UOG inventory, a 61% increase. Figure S2 shows the locations of UOG facilities in the Ft. McMurray AOSR area for the 2000 and 2011 UOG inventories. We can see that some UOG facilities that existed in 2000 have been closed while

many new facilities have opened since 2000. Updating the UOG inventory to the 2011-based 2013 projected inventory might thus be expected to have a significant impact on the spatial distribution of UOG emissions.

Given the availability of these new emissions data sets, the synthesized Phase 3 hybrid emissions inventory was created from the inventories listed in Table 4. As a complement to Table 2, which compared the VOC

emissions from the AOSR mines used for the three phases, Tables S3 to S5 compare the facility-total emissions of other CAC species compiled for the three phases from three main source types: CEMA off-road mobile emissions; facility smokestack and area-source emissions; and road-dust emissions. As described in the next section, further improvements were also made to the emissions processing methodology before new Phase 3 model-ready 2013 base-case emissions files were generated from the Phase 3 hybrid inventory. Additional

Phase 3 emissions input files that were generated for emissions sensitivity runs using an expanded set of CEMS measurements and aircraft-observation-based emissions estimates are then discussed in Section 4.

## 3 Phase 3 Emissions Processing for GEM-MACH 2013 Base-Case Simulations

The same overall emissions-processing methodology described in Zhang et al. (2015) and the JOSM report (ECCC & AEP, 2016) was used in Phase 3 to generate model-ready, gridded hourly emissions fields for GEM-

MACH using the SMOKE emissions processing system (https://www.cmascenter.org/smoke/). The three main steps required to process a typical emissions inventory that contains monthly or annual CAC emissions reported by jurisdiction for a small number of pollutants into hourly, gridded, model-ready emissions input files are (a) spatial disaggregation, (b) temporal disaggregation, and (c) chemical speciation (e.g., Dickson and Oliver, 1991; Houyoux et al., 2000; Moran et al., 2013b). Note that before spatial disaggregation (i.e., spatial allocation) can

be performed, a set of spatial surrogate fields must first be generated on the model grid of interest for such proxy or surrogate fields as population, road density, and agricultural land-use. Different inventories are then processed separately, often subinventory by subinventory (e.g., point sources, area sources, off-road sources, on-road sources), and as a last step some of the resulting gridded output fields may be merged.





Key aspects of the emissions-processing methodology for Phase 3 specific to the AOSR emissions included the following:

1) Facility-specific and process-specific spatial surrogate fields were again used (same as Phase 2) for the 10-km North American grid and 2.5-km western Canada grid based on GIS polygons of mine faces, tailings ponds, and plants for the six AOSR mining facilities (Figure 1) in order to spatially allocate the surface area emissions from off-road fleet and fugitive sources, between mine faces, tailings ponds, and plants. Emissions from individual smokestacks within these facilities, on the other hand, were treated as point-source emissions and assigned to the specific grid cells in which the stacks are located.

2) Facility-specific monthly temporal profiles for production-related emissions, such as emissions from off-road mine fleets and extraction plants, were generated based on facility-specific monthly production statistics for 2013 (Alberta Energy Regulator, 2014). Weekly and diurnal temporal profiles were treated as constant (i.e., "flat") as a default because the AOSR mining facilities usually operate around-the-clock throughout the year (note, however, the discussion on CEMS emissions in Section 4.1). Temperature-based monthly temporal profiles were created for fugitive VOC emissions from mine faces and tailing ponds, similar to the methodology that has been used in past AOSR environmental impact assessment (EIA) submissions (e.g., Cenovus, 2010; Imperial Oil, 2005).

3) Facility-specific and process-specific VOC speciation profiles were created based on VOC speciation profiles compiled in the CEMA inventory (Davies et al., 2012; Zhang et al., 2015).

4) PM speciation profiles from version 4.3 of the U.S. EPA SPECIATE database (https://www.epa.gov/air-emissions-modeling/speciate-version-45-through-40; Reff et al., 2009) were used to split PM emissions into six model chemical components: sulfate; nitrate; ammonium; elemental carbon; primary organic matter; and crustal material. Process-specific PM profiles were used for stack emissions based on the Source Classification Code (SCC) assigned to the stacks in the CEMA inventory (Davies et al., 2012). The "Unpaved Road" PM speciation profile from SPECIATE v4.3 was used to speciate fugitive dust emissions from unpaved roads within each facility in the base-case emissions.

Another required emissions processing step was to perform PM size disaggregation. As discussed in Makar et al. (2017, this issue) GEM-MACH may be configured to represent the PM size distribution with either two or 12 size bins. Accordingly, the PM emissions were processed twice, once for each PM size representation. The two-bin version separates $PM_{10}$ emissions into two size bins, $PM_{2.5}$ (fine bin) and PMC (coarse bin, equal to



PM$_{10}$ - PM$_{2.5}$), whereas the 12-bin version separates PM$_{10}$ emissions into the 10 size bins listed in Table 5, plus two larger size bins for diameters greater than 10 µm (note that the base-case emissions thus assumed no primary particulate emissions for sizes greater than 10 μm diameter).  For the 12-bin PM emissions, generic PM size distribution profiles were applied for three broad source types (area, mobile, and point) based on 10 source-

specific particle size distributions as discussed in Eldering and Cass (1996).  Figure 4 shows the distribution of the eight PM$_{2.5}$ bins for these three source types.  Mobile-source PM$_{2.5}$ emissions have a normal size distribution centred around 0.16 micron in diameter, but point-source and area-source PM$_{2.5}$ emissions are skewed to the smaller and larger size bins, respectively.

In addition to anthropogenic emissions, GEM-MACH must also consider natural emissions, including biogenic emissions, which depend on local vegetation type and light and/or temperature conditions.  GEM-MACH calculates biogenic emissions dynamically (that is, making use of the GEM meteorological model's predictions of temperatures and light levels during a simulation combined with vegetation-type-dependant biogenic emissions formulas from BEIS (Biogenic Emission Inventory System) v3.06).  Vegetation type is described

using the BELD3 (Biogenic Emissions Landuse Database, Version 3) database, which contains 230 vegetation classes at 1-km resolution (Pierce et al., 2000).  However, by 2013 the vegetation fields in the BELD3 database, which is based on early 1990's satellite imagery (Kinnee et al., 1997), were outdated over the AOSR mining area – much of the area which was forested in the 1990's but subsequently cleared of forest cover during the construction of the AOSR mining facilities.  This is illustrated in Figure 5, which shows mean Leaf Area Index

(LAI) for the gridded vegetation and corresponding summer peak isoprene emissions computed from the original BELD3 database.  Except for some areas within the two oldest AOSR mining facilities, Suncor Millenium/Steepbank and Syncrude Mildred Lake, LAI values and isoprene emissions over the other mining facilities as computed with the BELD3 database are erroneously high, due to the fact that these areas, which by 2013 had been cleared for surface mining, were still characterized in the database as forested.  Furthermore, the

only water bodies contained in the land cover database over this area are natural lakes.  The large artificial tailings ponds present in the mining facilities are not characterized as water-covered in the database (Figure 6a) even though in 2013, the tailings ponds in the AOSR covered an area of about 180 km$^2$ (http://www.energy.alberta.ca/OilSands/pdfs/FSTailings.pdf), the equivalent of 29 grid cells on the OS 2.5-km grid.  Tests of the GEM-MACH model's meteorology for plume-rise algorithm analysis have shown that these

artificial water bodies can have a significant influence on local meteorology and atmospheric vertical stability.  In addition, an examination of the default water-body field portion of the grid cells overlapping the Athabasca



River (centre of Figure 6a, flowing from south to north) showed that the river was also not being treated as a body of water in the default meteorological model database. The accuracy of the land-use database thus influences both meteorological and biogenic emissions estimation accuracy.

The outdated land-cover characteristics over the AOSR area would thus have an impact on GEM-MACH predictions, particularly at high spatial resolutions. To improve the land-use and vegetation characterization of this area, masks for cleared land and artificial water bodies were generated as GIS polygons based on 2013 satellite images. Rivers were added using more detailed GIS water-body data. By applying these masks to update vegetation and land-cover data, GEM-MACH-calculated biogenic emissions were significantly

impacted. Figure 7a shows the biogenic isoprene emissions over the AOSR surface mining area after the modification (cf. Figure 5b) and Figure 7b shows the difference between the original and modified isoprene emissions. The modified inland water coverage is shown in Figure 6b.

As an example of the emissions input files generated with the SMOKE emissions processing system from the

Phase 3 inventory, Figure S3 shows gridded August mean monthly emissions of six model pollutant species for a portion of the 2.5-km OS grid centred on the AOSR study area. Similar to Figure 7b, the locations of the six AOSR mining facilities can be seen clearly, but other emissions sources are also evident such as on-road vehicle emissions and emissions from the city of Fort McMurray. GEM-MACH results from the use of the new Phase 3 base-case emissions input files generated using these updated emissions inventories (Table 4), updated AOSR

facility-specific spatial surrogate fields, new monthly temporal profiles, new AOSR facility-specific VOC speciation profiles, and updated BELD3 vegetation and land-use data sets are described in Makar et al. (2017).

## 4 Additional Phase 3 Emissions Processing for GEM-MACH Sensitivity and Scenario Studies

In addition to the Phase 3 base-case emissions input files described in Section 3, additional GEM-MACH emissions input files were generated using four special emissions data sets in order to examine the effects of

specific changes to the emissions data on model predictions. These four data sets were (a) an expanded 2013 CEMS emissions data set, (b) 2013 OS field campaign aircraft-measurement-based VOC emissions estimates, (c) 2013 OS field campaign aircraft-measurement-based $PM_{2.5}$ emissions estimates, and (d) updated mercury emissions. These additional GEM-MACH emissions input files were used for a number of Phase 3 sensitivity studies that are described elsewhere in this special issue.



### 4.1 Expanded CEMS emissions data set

As noted in Section 2.1.2, CEMS-measured hourly $SO_2$ and $NO_x$ emissions from 17 stacks within four AOSR mining facilities were used in Phase 2 emissions processing for a GEM-MACH sensitivity test (ECCC & AEP, 2016; Makar et al., 2015; Zhang et al., 2015). This earlier work showed a relatively large impact of the better

time-resolved CEMS data on model results. Recall that in Canada, regulatory reporting at the national level requires only annual total emissions from large stacks; hence, details on specific time periods within the year are lost and calculations to reconstruct this time variation using each facility's operating schedule for the emitting activities can only be approximate. However, detailed CEMS records are reported to the Alberta provincial government. For Phase 3, CEMS measurements from about 100 stacks from 33 facilities with relatively large

$SO_2$ or NOx emissions were obtained for the province of Alberta for August and September, 2013. A sensitivity study was designed to investigate the impacts of both CEMS-measured hourly $SO_2$ and $NO_x$ emissions, and CEMS-measured stack volume flow rates and exit temperatures on GEM-MACH predictions (Akingunola et al., 2017b, this issue). For this study, the Phase 3 base-case stack emissions (based on 2013 NPRI annual reporting of stack emissions) were replaced with the corresponding CEMS hourly measurements. For the Phase 3 base-

case emissions, the stack flow rate and exit temperature, which are used to calculate plume rise, were assumed to be static at the annual reported values. However, CEMS-measured stack exit temperature and flow rate often display significant temporal variation as shown in Figure S4 for one example; hence, their values were generated in model-ready form for the two-month period to evaluate their impact on model predictions.

Due to the NPRI reporting threshold that facility operators are not required to report stack-specific emission from smokestacks shorter than 50 meters (Section 2.2), not all CEMS stacks could be matched to NPRI stacks. Overall, 38 of the 100 stacks in the expanded CEMS data set were matched with NPRI stacks at 20 facilities. However, since the 38 matched stacks were *de facto* all tall stacks with generally large emissions, emissions from the matched stacks account for 77% and 43% of total $SO_2$ and $NO_x$ emissions, respectively, from all NPRI

point sources in Alberta. Figures S5 and S6 show comparisons by facility of $SO_2$ and $NO_x$ emissions between the NPRI annual inventory and the two-month CEMS measurements for $SO_2$ and $NO_x$, scaled up to annual values. Overall, these scaled CEMS-based estimates agree well with NPRI annual totals, in spite of the large short-term temporal variation shown in the CEMS measurements. This is reasonable since facilities are expected to base their reported annual stack emissions on CEMS measurements. However, over shorter time

intervals the stack emissions levels may vary by up to several orders of magnitude, thus having a significant influence on model predictions. As well, the differences between CEMS volume flow rates and exit



temperatures and the annual reported values may also have a significant influence on model dispersion and transformation of emitted $SO_2$ and $NO_x$ (Akingunola et al., 2017a,b).

### 4.2 Aircraft-measurement-based VOC emissions estimates for AOSR mining facilities

As described in Li et al. (2017), aircraft observations of VOC species concentrations made during the 2013
AOSR field campaign have been used to estimate facility-total emissions of individual VOC species using a mass-balance approach (Gordon et al., 2015) for four AOSR mining facilities: Suncor Millenium/Steepbank; Syncrude Mildred Lake; Shell Canada Muskeg River/Jackpine; and CNRL Horizon (see Figure 1). Comparisons between the aircraft-observation-based estimates of individual VOC species emissions and the corresponding emissions reported to NPRI by these four facilities showed differences in terms of the magnitude
of both VOC species emissions and total VOC emissions (Li et al., 2017). To assess the impact of the suggested uncertainty of VOC emissions for these four facilities on GEM-MACH predictions, emissions of the individual VOC species estimated from the aircraft observations were mapped to the model VOC species used by GEM-MACH's ADOM-2 (Acid Deposition and Oxidant Model, version 2) gas-phase chemistry mechanism (Makar et al., 2003; Stroud et al., 2008) to replace the corresponding Phase 3 base-case model VOC species emissions for
the four facilities.

Table 6 shows a comparison of facility-total emissions of ADOM-2 model VOC species between the Phase 3 base-case emissions input files and the aircraft-observation-based emissions input files. Except for Syncrude Mildred Lake, the totals of the aircraft-observation-based VOC emissions for these facilities are higher than the
corresponding base-case totals, ranging from a factor of 2.5 for Suncor Millenium/Steepbank to 6.7 for Shell Canada Muskeg River/Jackpine and 7.2 for CNRL Horizon. The relative rankings of the emissions by model VOC species also differ for the two data sources. Figure 8 compares the process-specific VOC speciation profiles for these four facilities that were used for the Phase 3 base-case study based on the CEMA inventory (Davies et al., 2012; Zhang et al., 2015). Figure 8 also compares the inventory-based VOC speciation profiles
with the aircraft-observation-based VOC speciation profiles by facility. As the emissions estimates from the aircraft observations corresponded to facility-total emissions, an emissions-weighted, base-case "composite" VOC speciation profile was created for each facility by combining the plant, mine-face, and tailings-pond VOC speciation profiles based on the emissions of each ADOM-2 model VOC species. Both the aircraft-observation-based VOC speciation profiles and the "composite" VOC profiles vary from facility to facility, but there are
some differences between the two profile types. Consistent with Li et al. (2017), for example, the aircraft-



observation-based VOC profiles have a higher propane emissions fraction and a much lower higher-aromatic emissions fraction than the composite profiles for all four facilities. The aircraft also measured significant amounts of isoprene emissions from the Suncor Millenium/Steepbank and the CNRL Horizon facilities, which are not present in the corresponding base-case profiles.

Figure S7 shows spatial variations in the ratio of the gridded, model-ready, aircraft-observation-based higher-alkane emissions to corresponding base-case emissions for the GEM-MACH 2.5-km grid over the AOSR study area. Consistent with Table 6, the ADOM-2 higher-alkane emissions estimated from aircraft observations are about eight times higher for the Shell Canada Muskeg River/Jackpine and CNRL Horizon facilities than

corresponding emissions from the 2013 NPRI but are closer for the Suncor Millenium/Steepbank and Syncrude Mildred Lake facilities. The variations seen within individual facilities are due to different emission rates for plants, mine faces, and tailings ponds. As expected there is no difference for areas outside of these four facilities. The new GEM-MACH emissions input files generated using the aircraft-observation-based VOC emissions have been used for a GEM-MACH sensitivity test (see Stroud et al., 2017, this issue).

**4.3 Aircraft-measurement-based PM emissions estimates for AOSR mining facilities**

PM emissions from the AOSR mining facilities originate mainly from four major source categories: (1) emissions from plant stacks; (2) tail-pipe emissions from the off-road mining fleet; (3) fugitive dust originating from various activities, such as excavation of oil-sand ore, loading and unloading trucks, and wheel abrasion of surfaces by off-road vehicles; and (4) wind-blown dust. As summarized in Table 4, PM emissions

from plant stacks and fugitive dust source categories were obtained from the 2013 NPRI while emissions from tail-pipe emissions were provided by the 2009/10 CEMA inventory. However, none of the inventories included wind-blown dust emissions, and the estimates of anthropogenic fugitive dust emissions are highly uncertain. In addition, emissions of construction dust from one facility, the Imperial Oil Kearl mine, which was still under construction during the aircraft monitoring campaign, were expected to be large. In order to evaluate and

potentially to improve these emissions estimates, estimates of size-resolved PM emissions were also calculated based on aircraft measurements of size-resolved PM concentrations made during the 2013 AOSR field campaign for all six AOSR mining facilities.

The 2013 aircraft campaign used a top-down mass balance approach (Gordon et al., 2015) to determine PM

emissions from all six AOSR surface mining facilities. For particles with a diameter in the range of 0.3 to 20





µm, a Forward Scattering Spectrometer Probe (FSSP) model 300 was deployed from a wing-mounted pod (Baumgardner et al., 1989) to measure the particle number concentration size distribution in 30 size bins. An Ultra-High Sensitivity Aerosol Spectrometer (UHSAS) was used inboard to determine the number concentration size distribution of particles with diameter from 0.06 to 1.00 µm in 99 size bins. Volume concentration size

distributions of particles were derived from these number concentration size distributions from 0.06 to 20 µm by combining both sets of measurements from the two instruments. Size-dependent particle densities, varying from 1.5 to 2.5 g/cm$^3$, were used to convert the volume concentration size distributions to mass concentration size distributions, based on the known mineralogy for the supermicron particles for the top soil in the region and the known chemical composition for submicron particles from concurrent aerosol mass spectrometer measurements

(Liggio et al., 2016). The resulting particle mass concentration size distributions were combined to match the 12-bin version of the GEM-MACH model particle size distribution. The mass balance emission algorithm TERRA (Top-Down Emission Rate Retrieval Algorithm) (Gordon et al., 2015) was then applied to these particle size bins to determine the particle mass emission rates for each bin. Uncertainties in the particle mass emission rate from each facility determined this way were estimated at approximately 36% for supermicron

particles, and 32% for submicron particles. Based on the aircraft observations, 68% of the $PM_{10}$ emissions are in the coarse mode (2.5 to 10 µm).

Figure 9 shows a comparison of annual facility-level $PM_{2.5}$ emissions between the base-case inventory-based values and the aircraft-observation-based estimates for the six AOSR facilities. Note that the latter were

annualized for this comparison simply by assuming constant daily emissions throughout the year, which does not account for modulation by snow cover, frozen ground, or precipitation, but the aircraft-observation-based estimates were only used in GEM-MACH for summertime modelling. Except for the Imperial Oil Kearl facility, the $PM_{2.5}$ emissions estimated from aircraft observations were at least one order of magnitude larger than the 2013 APEI $PM_{2.5}$ emissions used for the Phase 3 base-case emissions processing. One reason for the

difference is that wind-blown dust is not included in the inventory, which was compiled for anthropogenic emissions only. For the base case, total $PM_{2.5}$ emissions from off-road vehicle tail-pipe emissions and stacks are 2,272 tonnes/year (Tables S3 and S4) while road dust emissions are 4,134 tonnes/year (Table S5). Thus, anthropogenic fugitive dust emissions account for 65% of total $PM_{2.5}$ emissions from the AOSR mines. Aircraft-observation-based estimated total $PM_{2.5}$ emissions from all six facilities are about 61,500

tonnes/year. If we assume that all of the unreported $PM_{2.5}$ emissions come from natural wind-blown dust, then fugitive dust accounts for 96% of total $PM_{2.5}$ emissions from those facilities.





Figure 10 shows the observed size distribution of the first eight GEM-MACH size bins, which correspond to the PM$_{2.5}$ size range (cf. Table 5). Although the size distribution of the PM$_{2.5}$ emissions varies from facility to facility, 65%–95% of PM$_{2.5}$ emissions are in Bin 8 (diameter range from 1.28 to 2.56 μm), implying that the

majority of the PM$_{2.5}$ emissions are from fugitive-dust area sources, either from dust kicked up by off-road mining vehicles or from wind-blown dust. Compared to the area-source PM size distribution profile used by SMOKE to process the base-case emissions (Figure 4), a much larger Bin 8 mass fraction and smaller Bin 1 to 7 (i.e., <1.28 μm) mass fractions were observed by the aircraft for the AOSR mining facilities.

An AOSR-specific PM chemical speciation profile consisting of six chemical components was also constructed for fugitive dust emissions from these facilities to replace the standard "Fugitive Dust" profile from the U.S. EPA SPECIATE v4.3 database (see Section 3). Wang et al. (2015) analysed soil samples collected from 17 AOSR facility sites and 10 forest sites. The samples were further characterized as paved road dust, unpaved road dust, tailings sands, and overburden soil. Their analysis showed that PM speciation is clearly different

between the dust collected from the facility sites and from the forest sites. For this study, the new AOSR-specific fugitive-dust PM speciation profile was compiled by averaging the site-specific profiles from all 17 facility sites from Wang et al. (2015) to represent surface PM speciation with the following three exceptions:

1) For the unpaved-road site S16, the elemental-carbon percentage seemed to be too large, which might be an artefact due to dry deposition from heavy-duty diesel exhaust (Wang et al., 2015). This site was

excluded from the facility profile average in their study and was excluded in this study too.

2) The organic-carbon percentage for site S10 was much smaller and the elemental-carbon percentage was larger than those of other facility sites. That site was excluded from the organic-carbon range discussion in Wang et al. (2015) and was excluded here as well.

3) S17 is located on Highway 63, so it was also excluded from the facility average.

Figure 11 shows a comparison of the fugitive-dust PM speciation profile used for the Phase 3 base-case emissions processing, which is the standard "Unpaved Road" profile from the U.S. EPA SPECIATE v4.3 database, and the new profile described above. The organic-matter (OM) percentage in the AOSR-specific PM speciation profile (21.8%) is about three times larger than the fraction in the standard "Unpaved Road" profile

(7.6%), suggesting that soils in the AOSR facilities contain more organic matter than soils in other areas. The



crustal-material percentage decreases correspondingly, from over 91% to 76%. The AOSR-specific PM speciation profile also has more sulfate and elemental carbon, but the fractions are relatively small.

Figure S8 shows spatial variations in the ratio of the gridded aircraft-observation-based Bin 8 OM emissions to
the corresponding base-case emissions for the GEM-MACH 2.5-km grid over the AOSR study area. Except for the Imperial Oil Kearl facility, OM emissions estimated from the aircraft observations are more than two orders of magnitude larger than those for base-case study due to the combination of higher PM emissions (Figure 9), larger Bin 8 mass fraction (Figures 4 and 10), and the larger OM mass fraction (Figure 11).

The new estimates of total fugitive dust emissions and the new PM size-distribution and speciation profiles were used for two GEM-MACH sensitivity simulations. One of these simulations focussed on the impact of the increases of VOC and primary OM emissions on total organic aerosol and the formation of secondary organic aerosol (Stroud et al., 2017, this special issue). The second examined the impact of the increased crustal-material emissions on acid deposition by making use of the Wang et al. (2015) PM speciation profile to further
speciate the model's crustal material into a base-cation fraction (Makar et al., 2017, this special issue). Both studies suggest that these improvements to emissions have a significant impact on model performance. The acid deposition study (Makar et al., 2017) also found that the fugitive dust estimates from the 2013 aircraft field study, while larger than the reported inventory values, may themselves underestimate the total fugitive dust emissions when compared to deposition observations in the immediate vicinity of the oil sands.

**4.4 Mercury emissions**

Mercury emissions from the SMOKE-ready versions of the 2010 Canadian APEI and version 1 of the 2011 U.S. NEI (NEIv1) were used in Phase 2 for creating gridded GEM-MACH-ready mercury emissions. In Phase 3 these emissions input files were updated with two AOSR-specific adjustments. First, annual total mercury emissions to air from all NPRI facilities in the 2010 Canadian APEI, including the six AOSR mining facilities,
were 3,429 kg/year. In comparison, the annual total mercury emissions to air reported by all NPRI facilities for 2013 were 2,529 kg/year. Thus for the 2013 field study, the 2013 NPRI reported values were used for the model Hg emissions. Second, the U.S., mercury emissions from off-road vehicles were only available for the state of California in the SMOKE-ready version of the 2011 NEIv1 (https://www.epa.gov/air-emissions-modeling/2011-version-6-air-emissions-modeling-platforms), whereas the original 2011 NEIv1 (https://www.epa.gov/air-
emissions-inventories/2011-national-emissions-inventory-nei-data) included off-road-mobile mercury emissions



for other states as well. The amount of off-road-mobile mercury emissions for California was the same in the two inventory versions. Based on the original 2011 NEIv1 inventory, total annual off-road-mobile mercury emissions for the entire U.S. were 40.9 kg/year, of which 26.1 kg/year was from California. Although these off-road-mobile mercury emissions were relatively small compared with other emissions sources (see Table 7) and

more than 60% of the off-road-mobile mercury emissions were from California, the second adjustment was to use off-road-mobile mercury emissions from the original 2011 NEIv1 to add in mercury emissions for the missing states in the off-road-mobile subinventory of the SMOKE-ready version of the 2011 U.S. NEIv1.

Table 7 presents a summary of source-specific anthropogenic mercury emissions used for Phase 3 for both the

U.S. and Canada. Total 2011 U.S. annual mercury emissions from all four broad categories were 46,992 kg, of which nearly 90% was from point sources and the rest were mainly from area sources (9%). Mercury emissions from on-road and off-road vehicles accounted for less than 1% of total mercury emissions, and most of these vehicular emissions (90%) came from on-road vehicles. The summary of 2010/2013 Canadian mercury emissions shows that point sources were the largest anthropogenic source of mercury emissions in Canada

(58%), followed by area sources (42%), and on-road and off-road vehicle emissions contributed little. Total mercury emissions from Canada for 2010/2013 were about 9% of those emitted in the U.S. for 2011. The two adjustments made for Phase 3 reduced U.S. and Canadian anthropogenic mercury emissions by 885 kg/year or less than 2%. However, emissions of mercury from forest fires were also recognized as a major source (Fraser et al., 2017, this special issue).

Three mercury species (elemental, divalent gas, and particulate) are considered in the mercury version of the GEM-MACH model (Fraser et al, 2017). Mercury emissions for the Canadian 2013 NPRI point source emissions were pre-speciated based on the 2006 Canadian point-source emissions inventory used for the 2008 mercury assessment (UNEP, 2008). For other inventories, mercury emissions were reported as unspeciated

totals in the 2010 Canadian APEI and the 2011 U.S. NEIv1. For these other inventories, mercury speciation was carried out using speciation profiles for nine broad source categories following the same methodology used in the U.S. EPA 2005 NEIv4.1 platform. The same profiles had also been used in the U.S. EPA 2002v3 platform (see Table 3-14 in U.S. EPA, 2011).

Figure 12 shows the spatial distribution of Phase 3 elemental mercury emissions for both Canada and the U.S. on the 10-km GEM-MACH continental grid for one afternoon hour in August. Most of the mercury emissions



are from populated and industrial areas. Figure S9 shows the domain-average percentages of the three mercury species based on total emissions summed over the nine source categories. About 50%, 30%, and 20% of the total mercury emissions are in the elemental, divalent gas, and particulate states, respectively. Fraser et al. (2017, this issue) present some results from the use of these Phase 3 mercury emissions input files.

## 5 Summary and Future Work

A number of sets of model-ready emissions input files have been prepared over the past six years in three phases for the GEM-MACH air quality modelling system in support of the Governments of Canada and Alberta Joint Oil Sands Monitoring (JOSM) plan. These emissions files were used by GEM-MACH to conduct nested AQ forecasts in support of an Oil Sands field campaign carried out in summer 2013 as well as ongoing experimental forecasts since then and retrospective model simulations and analyses for the field-study period. Two GEM-MACH grids were considered: a North American continental grid with 10-km grid spacing and a high-resolution western Canada grid with 2.5-km grid spacing centred over the Athabasca Oil Sands Region (AOSR) of north-eastern Alberta, Canada.

Successful preparation of emissions input files for AQ modelling requires accurate and representative emissions inventories and emissions processing. The JOSM Phase 1 emissions processing undertaken from 2012 to 2013 was a particular challenge because the base years of all available emissions inventories were three years or more out of date compared to 2013, that is, 2010 or earlier. Moreover, the six large AOSR mining facilities that were the focus of the 2013 field campaign were changing year by year, which made emissions representativeness an important issue. These facilities are also complex and unconventional industrial sources that cannot be well represented by standard emissions processing approaches.

The approach adopted in Phase 1 was to review all available emissions inventories covering the study area and to extract the best available information from the 10 inventories considered in order to construct a detailed synthesized or hybrid AQ modellers' emissions inventory for this specific project. One important change in Phase 1 to the emissions processing methodology that was used with the new hybrid modellers' inventory was to treat three types of major emissions sources within each AOSR mining facility – mine faces, tailings ponds, and extraction plants – as area sources rather than point sources due to their large spatial extent. This required



three spatial surrogate fields to be developed for each individual facility based on a 2010 GIS shapefile describing the AOSR mines. Chemical speciation profiles were also chosen to be as source-specific as possible

For Phase 2 emissions processing from 2014 to 2015, more up-to-date emissions inventories and other relevant

emissions information became available, including a modellers' version of a newer (2010) Canadian national comprehensive emissions inventory (APEI), a preliminary version of the 2013 Canadian large-point-source inventory (from NPRI), monthly bitumen production statistics for 2013 that included a new AOSR mining facility (Imperial Oil Kearl), continuous emissions monitoring system (CEMS) data sets for 2013 for 17 smokestacks in four AOSR mining facilities, and updated, 2013-specific AOSR shapefiles. A more

comprehensive and detailed description of the modellers' inventory compilation and emissions processing for Phases 1 and 2 to prepare GEM-MACH emissions input files is contained in a JOSM report (ECCC & AEP, 2016), which also identified gaps and recommended areas for future improvements.

This paper focused on the Phase 3 emissions processing that was carried out from 2016 to 2017. Some of the

gaps and recommendations raised in the JOSM report were addressed during this phase. Canadian area and point source emissions were updated to a 2013 criteria air contaminant (CAC) inventory (the AQ modellers' version of the 2013 APEI), which included a new upstream oil and gas subinventory, and U.S. emissions were updated to version 1 of the 2011 NEI to replace an earlier projected 2012 NEI. An expanded CEMS data set of hourly $SO_2$ and $NO_x$ emissions and smokestack operating characteristics for August–September 2013 was

obtained for the entire province of Alberta, increasing the provincial total coverage of point source $SO_2$ and $NO_x$ emissions by CEMS measurements from 31% and 3% to 77% and 43%. New VOC and PM emissions estimates and chemical speciation profiles for the AOSR mining facilities that had been derived from on-site surface observations and aircraft observations made during the 2013 field campaign were processed for several GEM-MACH sensitivity studies. The aircraft-observation-based VOC emissions were about two times larger than the

base-case emissions from the 2013 NPRI (Li et al., 2017). For PM emissions, a comparison between the annualized aircraft-observation-based emissions and the NPRI annual emissions shows a factor-of-10 difference (Figure 9). The VOC and PM chemical speciation profiles used to speciate emissions from the AOSR mines were also noticeably different than those used to process the Phase 3 base-case emissions. A vegetation data base used to estimate biogenic emissions and a land-cover data base used in the parameterizations of land-

surface processes and dry deposition were also modified to account for the rapid change of vegetation cover and land use in the AOSR region due to year-by-year changes in surface mining activities. In addition to CAC





emissions, mercury emissions were also processed to support mercury modelling activities using newly available data sets.

The various Phase 3 emissions input data sets have been used to drive a number of GEM-MACH simulations as well as to evaluate plume rise algorithms, results from which are discussed in a number of companion papers in this special issue: see Akingunola et al., 2017b; Fraser et al., 2017; Gordon et al., 2017, Makar et al., 2017; Russell et al., 2017; Stroud et al., 2017, and Whaley et al., 2017.

This study also provides specific examples of some common issues related to the preparation of emissions input files for AQ models. First, there is always a time lag between a year of interest and the year in which an emissions inventory becomes available for that year of interest. Second, inventories are always subject to change due to reported corrections or to changes in estimation methodology. Third, if multiple inventories are available for the same region and the same base year, they are unlikely to be in perfect agreement. Fourth, a synthesized or hybrid inventory can provide a more accurate representation of emissions than any of its component inventories. And fifth, extra effort and investigation related to the specific year and region of interest can yield significant improvements over standard emissions processing methodologies.

Nevertheless, although improved sets of emissions input files were generated during Phase 3 after a considerable effort to acquire and apply new sources of emissions data representative of the 2013 AOSR field-study period, there are still large uncertainties associated with these emissions. Here are six areas that still need further improvement:

1) Aircraft measurements made in late summer 2013 during the AOSR field study show that VOC and PM emissions reported to the NPRI using currently accepted estimation methods are underestimated for the AOSR facilities (Li et al., 2017). However, these measurements were made during a limited time period (four weeks) and the mass-balance calculations used to estimate emissions were only applied to a relatively large area (Gordon et al., 2015; Li et al., 2017). Large variations in PM emissions results were also seen from flight to flight for the same facilities, probably related at least in part to the variation of mined volume of oil sands from day to day or recent precipitation. There are thus still issues with the spatial and temporal allocation of emissions to the right location at the right time. More aircraft measurements, especially at other times of year, and further attempts to spatially



locate emissions on a sub-facility level, are needed to confirm and augment the findings of the 2013 field study.

5 The aircraft measurements also indicated that the VOC speciation reported to NPRI by individual AOSR mining facilities needs to be improved (Li et al., 2017). Moreover, these aircraft measurements were carried out at the facility level, but within these very large facilities the individual VOC species emitted from mine faces, tailings ponds, and plants can be very different. Additional measurements of emissions at the sub-facility level, from mine faces, tailings ponds, and plants for multiple AOSR facilities are needed to further improve emissions factors, temporal profiles, and chemical speciation 10 profiles that can improve the emissions inventory and emissions processing (e.g., Small et al., 2015; Stantec Consulting Ltd. et al., 2016). Given the above differences between field study measurements and reports, the AOSR mining facilities should also review the methodologies that they employ to estimate and report VOC emissions to NPRI.

15 2) The off-road mining fleets in the six AOSR mining facilities are a large source of NOx emissions, but large differences are seen in the emissions estimates for this source sector between different inventories. For example, the 2010 CEMA inventory lists 38,362 tonnes of NOx emissions for this sector, but the 2010 APEI for the same year lists 27,786 tonnes. The 2013 APEI then reduced NOx emissions from the OS off-road mining fleets to 12,370 tonnes. Since  mined oil sands increased by 20 17% between 2010 to 2013, the significant drop of NOx emissions is probably due to different emissions factors being used for these two inventory years (possibly due in part to the introduction of cleaner heavy-hauler trucks: e.g., M.J. Bradley & Associates, 2008).

Additional sources of information are needed to reconcile the differences amongst existing inventories. 25 One possible data source is satellite remote sensing. For example, a methodology has been developed recently to use repeated satellite measurements of $NO_2$ vertical column density over the AOSR to estimate NOx emissions (McLinden et al., 2014, 2016). Preliminary results show that area source NOx emissions in the OS area, which are mainly from the off-road fleets, are about 38kt per year for 2013 comparable to the 2010 CEMA inventory. The 2010 CEMA inventory was also deemed to have the 30 best estimation of off-road emissions for the AOSR facilities (ECCC & AEP, 2016). Satellite remote sensing (e.g., McLinden et al., 2014; Shephard et al., 2015; Sioris et al., 2017) and ground-based



remote sensing (e.g., Fioletov et al., 2016), should thus be considered in future for emissions estimation and verification.

3) There have been ongoing efforts to improve the spatial allocation of emissions within the huge AOSR mining facilities using spatial surrogate fields generated from the locations of mine faces, tailings ponds, and extraction/upgrading plants. For example, the 2010 version of the shapefile used for generating these surrogates was updated in Phase 2 based on 2013 satellite images (Zhang et al., 2015). Further improvements, however, are possible. As one example, the spatial surrogate used to allocate emissions from the off-road mining fleet currently allocates all of the emissions to the mine-face locations and does not account for the movement of the heavy-hauler trucks between the mine faces and the extraction plants. Year-specific shapefiles with locations of active mining areas and current boundaries of tailing ponds as well as activity data sets for the actual or average movement of mining vehicles and time spent at locations throughout the mine should be obtained to improve the spatial allocation of off-road emissions for the AOSR mining operations (ECCC & AEP, 2016)

4) Fugitive VOC emissions from tailing ponds and mine faces are currently provided as annual totals in the inventory. A temperature-based monthly temporal profile was used to allocate the annual emissions to each month while weekly and diurnal temporal profiles were assumed to be constant, which is likely not realistic. For example, night-time emission rates over the mining faces are likely lower than daytime rates due to lower surface temperatures. In future, model-predicted or locally measured hourly temperature and wind speed may be used to estimate hourly fugitive VOC emissions if the dependence of fugitive VOC emission rates on temperature and wind speed can be parameterized (Li et al., 2017). A related issue is that tailings ponds are of different ages; some are receiving fresh tailings while others have been inactive for years, which may mean lower emission rates due to past off-gassing of more volatile components. Consideration should thus be given to tailings-pond age when allocating VOC emissions between different tailings ponds. A recently completed study (summer 2017) of tailings pond emissions is expected to lead to improved estimates of emissions from these sources.

5) Fugitive dust emissions estimates based on aircraft observations suggest large underestimates in the reported inventory totals, and GEM-MACH modelling suggests that even these revised estimates, or



the fraction of their mass which is composed of base cations, are underestimated (Makar et al., 2017, this issue). Further aircraft-based measurements of fugitive dust emissions and their speciation are needed to improve the emissions inventories used here. A parameterization of wind-blown dust emissions should also be added to GEM-MACH.

6) For mercury emissions, although unspeciated mercury emissions were obtained from inventories with base years close to 2013, chemical speciation was done crudely using speciation profiles for nine broad source categories. This methodology needs to be updated as more detailed speciation information becomes available in the future.

**ACKNOWLEDGEMENTS**

Emissions inventories used in this study were provided by the Pollutant Inventory and Reporting Division of ECCC, the Cumulative Environmental Management Association, and the U.S. Environmental Protection Agency. The Alberta CEMS data were provided by Marilyn Albert, Ewa Przybylo-Komar, Katelyn Mackay, and Tara-Lynn Carmody of Data Management and Stewardship, Corporate Services Division, Alberta
Environment and Parks. The project was supported by the Climate Change and Air Quality Program (CCAP) and the Oil Sands Monitoring program (OSM). We also appreciate the information provided by Sunny Cho and Richard Melick of Albert Environment and Parks about emissions in the AOSR and the province of Alberta. Finally, we thank our colleagues in the Program Integration Division of ECCC for their careful internal review of the manuscript.

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



**Table 1: Summary of Canadian emission sources used for generating JOSM Phase 1 emissions input files.**

| Data Category | Data Sources |
|---|---|
| Point/Facility Sources | • 2009/10 CEMA Inventory for AOSR study area (except VOC, NH$_3$, PM$_{10}$) |
| | • 2010 NPRI for rest of the domain |
| OS Off-road Fleet | • 2009/10 CEMA Inventory |
| Fugitive Dust from Major Facility | • 2010 NPRI |
| Tailings Ponds, Mines and Plant Fugitives | • 2010 facility-total VOC emissions from CEMA scaled by NPRI:CEMA |
| | • Splitting factors for fugitive VOC emissions for tailings ponds, mines and plants based on 2009/10 CEMA Inventory |
| Small & Medium Upstream Oil and Gas (UOG) Sources | • 2006 APEI (projected to 2006 from the 2000 Canadian upstream oil and gas emissions inventory) |
| Non-Mobile Area Sources | • 2006 APEI |
| Mobile Sources | • 2006 APEI |

**Table 2: Comparison of annual facility-total VOC emissions (tonnes) between 2010 NPRI, 2010 CEMA, and versions 1 and 2 of the 2013 NPRI for the OS mining facilities within the AOSR study area.**

| Emissions Processing Phase | | | 1/2 | | 3 |
|---|---|---|---|---|---|
| **Facility Name** | **2010 APEI/NPRI** | **Original 2010 CEMA** | **2010-NPRI-Scaled CEMA** | **2013 APEI/NPRI** | **2013 NPRI version 2** |
| Suncor Millenium/Steepbank | 28,940 | 10,808 | 28,013 | 6,768 | 9,529 |
| Syncrude Mildred Lake | 8,591 | 7,663 | 19,861 | 8,291 | 20,732 |
| Syncrude Aurora North | 5,182 | 3,319 | 8,602 | 2,572 | 8,268 |
| Shell Muskeg River/Jackpine | 1,460 | 2,813 | 7,291 | 2,614 | 2,614 |
| CNRL Horizon | 27,853 | 2,623 | 6,798 | 4,328 | 4,560 |
| Imperial Oil Kearl | | | | 2,546 | 2,546 |
| **Total** | **72,026** | **27,226** | **70,566** | **27,119** | **48,249** |



**Table 3:** **Summary of Canadian emission sources used for generating JOSM Phase 2 emissions input files.**

| Data Category | Data Sources |
|---|---|
| Point/Facility Sources | • 2009/10 CEMA Inventory for AOSR study area (except VOC, NH$_3$, PM$_{10}$) |
| | • 2010 NPRI for rest of the domain |
| | • 2013 preliminary NPRI for AOSR Imperial Kearl facility and for NH$_3$ emissions |
| | • SO$_2$ and NO$_x$ from CEMS measurements for stacks of the OS facilities during study period |
| | • SO$_2$ from CNRL daily reports during one-week period in August 2013 |
| OS Off-road Fleet | • 2009/10 CEMA Inventory |
| Fugitive Dust from Major Facility | • 2013 preliminary NPRI |
| Tailings Ponds, Mines and Plant Fugitives | • 2010 facility-total VOC emissions from CEMA scaled by NPRI:CEMA |
| | • Splitting factors for fugitive VOC emissions for tailings ponds, mines and plants based on 2009/10 CEMA Inventory |
| Small & Medium UOG Sources | • 2010 APEI (projected to 2010 from the 2000 Canadian UOG emissions inventory) |
| Non-Mobile Area Sources | • 2010 APEI |
| Mobile Sources | • 2010 APEI |

**Table 4:** **Summary of Canadian data sources used for generating JOSM Phase 3 base-case emissions input files.**

| Data Category | Data Sources |
|---|---|
| Point/Facility Sources | • 2013 NPRI v1 for the whole domain except for the OS facilities |
| | • 2013 NPRI v2 for the OS facilities, but 2009/2010 CEMA stack information used |
| OS Off-road Fleet | • 2009/10 CEMA Inventory |
| Fugitive Dust from Major Facility | • 2013 NPRI v1 |
| Tailings Ponds, Mines and Plant Fugitives | • Facility-total VOC emissions from 2013 NPRI v2 |
| | • Splitting factors for fugitive VOC emissions from tailings ponds, mines and plants based on 2009/10 CEMA Inventory |
| Small & Medium UOG Sources | • 2013 APEI (projected from the 2011 Canadian UOG inventory) |
| Non-Mobile Area Sources | • 2013 APEI |
| On-road and Off-road Mobile Sources | • 2010 APEI |



**Table 5: PM10 size-bin ranges as Stokes diameter (µm) for the 12-bin version of GEM-MACH.**

| Bin 1 | Bin 2 | Bin 3 | Bin 4 | Bin 5 | Bin 6 | Bin 7 | Bin 8 | Bin 9 | Bin 10 |
|-------|-------|-------|-------|-------|-------|-------|-------|-------|--------|
| 0.01- | 0.02- | 0.04- | 0.08- | 0.16- | 0.32- | 0.64- | 1.28- | 2.56- | 5.12- |
| 0.02 | 0.04 | 0.08 | 0.16 | 0.32 | 0.64 | 1.28 | 2.56 | 5.12 | 10.24 |

**Table 6: Comparison of speciated annual ADOM-2 model VOC species emissions (tonnes/year) between base-case emissions from the 2013 NPRI version 2 and the aircraft-observation-based estimates. Note that unknown or unreactive VOC species are not included.**

| | Suncor – M/S | | Syncrude - ML | | Shell – MR/J | | CNRL - Horizon | |
|---|---|---|---|---|---|---|---|---|
| **SPECIES** | **Base Case** | **Aircraft** | **Base Case** | **Aircraft** | **Base Case** | **Aircraft** | **Base Case** | **Aircraft** |
| Higher Alkenes | 601 | 1,038 | 863 | 513 | 34 | 1,219 | 177 | 1,657 |
| Higher Alkanes | 5,636 | 13,488 | 12,348 | 10,022 | 1,690 | 14,384 | 2,651 | 23,779 |
| Higher Aldehydes | 15 | 0.0 | 40 | 301 | 64 | 28 | 10 | 0.0 |
| Higher Aromatics | 1,457 | 1,569 | 5,273 | 1,696 | 746 | 88 | 1,125 | 500 |
| Propane | 0.5 | 953 | 0.0 | 1,592 | 3.1 | 955 | 0.0 | 1,928 |
| Ethene | 8.0 | 0.0 | 15 | 77 | 0.2 | 290 | 3.5 | 0.0 |
| Formaldehyde | 3.8 | 235 | 4.5 | 647 | 0.7 | 0.0 | 0.7 | 0.0 |
| Isoprene | 0.3 | 2,230 | 0.5 | 0.0 | 0.3 | 143 | 0.1 | 1,346 |
| Toluene | 486 | 1,112 | 806 | 1,539 | 6.8 | 72 | 135 | 393 |
| Methyl ethyl ketone | 0.0 | 0.0 | 0.0 | 212 | 0.0 | 0.0 | 0.0 | 0.0 |
| **Total VOC** | **8,208** | **20,625** | **19,350** | **16,600** | **2,545** | **17,180** | **4,102** | **29,603** |

**Table 7: Sum of source-sector-specific mercury emissions (kg) for the 2011 U.S. inventory (version 1) and the 2010/2013 Canadian inventory.**

| Source Category | 2011 U.S. | 2010/13 Canada |
|---|---|---|
| **Point** | 42,202 | 2,529 |
| **Area** | 4,321 | 1,803 |
| **On-road** | 358 | 2.3 |
| **Off-road** | 41 | 0.0 |
| **Total** | **46,922** | **4,334** |



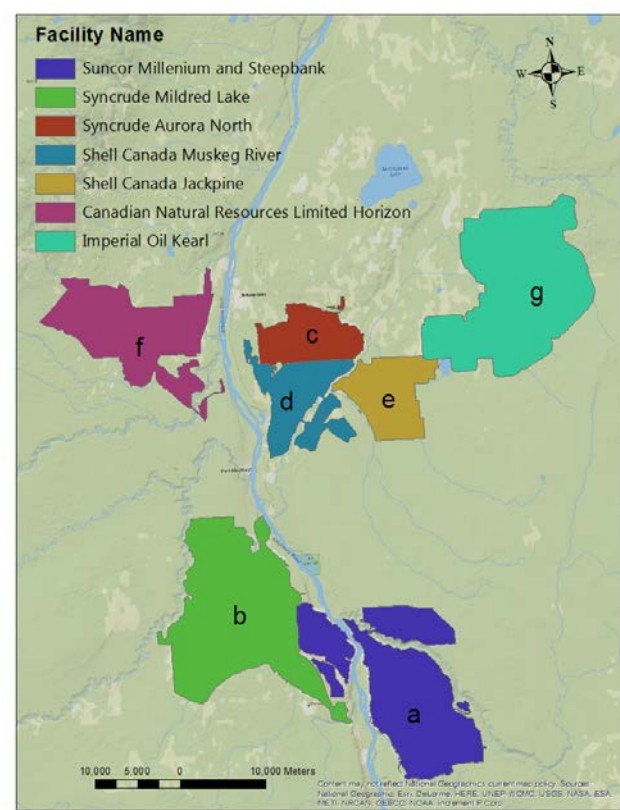

20   **Figure 1:   Location of six AOSR surface mining and processing facilities:   (a) Suncor Millenium and Steepbank; (b) Syncrude Mildred Lake; (c) Syncrude Aurora North; (d) Shell Canada Muskeg River and (e) Shell Canada Jackpine (reported to NPRI as one facility); (f) Canadian Natural Resources Limited Horizon; and (g) Imperial Oil Kearl (only started production in 2013, not considered in earlier inventories). The city of Fort McMurray is located about 10 km to the south.**



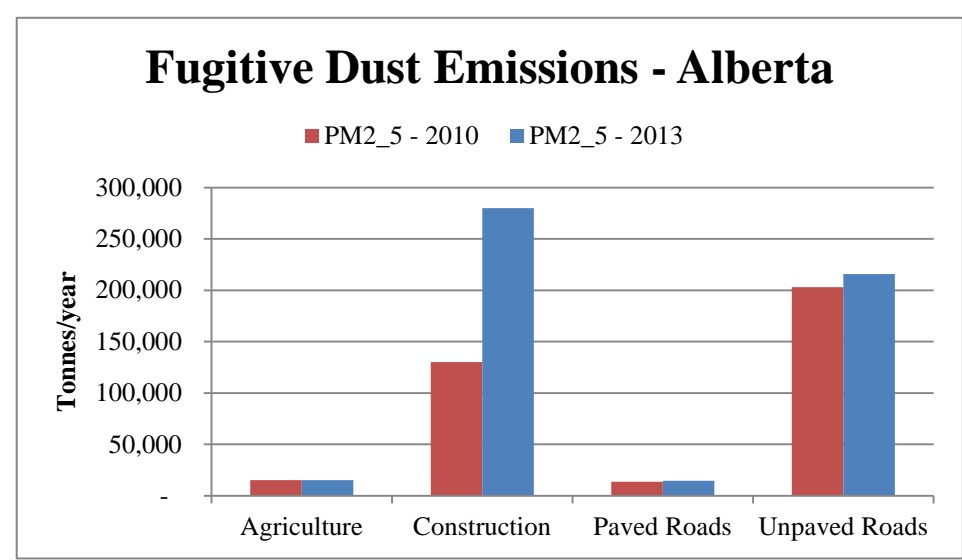

**Figure 2: Comparison of fugitive PM$_{2.5}$ emissions for four sectors between 2010 APEI (used for Phase 2) and 2013 APEI (used for Phase 3) for the province of Alberta.**

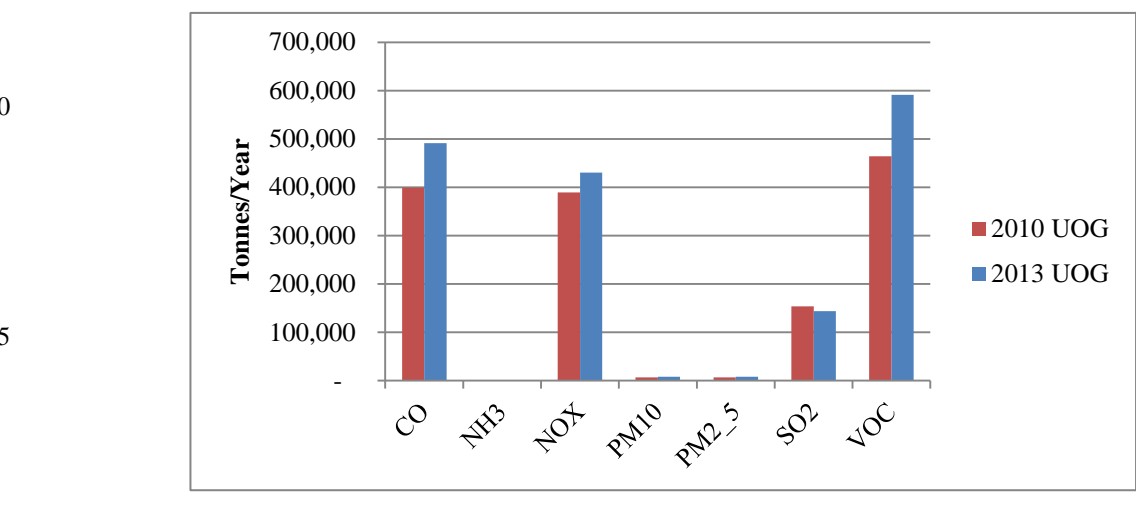

**Figure 3: Comparison of national CAC emissions between the year-2000-based projected 2010 UOG inventory and the year-2011-based projected 2013 UOG inventory.**





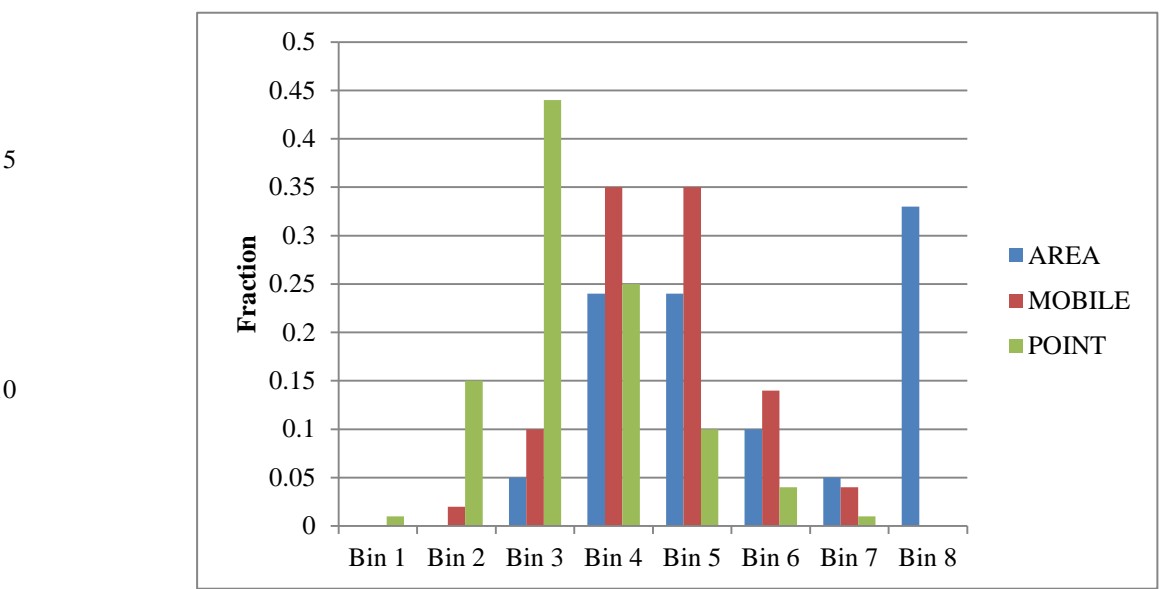

15  **Figure 4:  Fractional distribution of the eight PM$_{2.5}$ size bins for the 12-bin version of GEM-MACH modelling for three broad types of emissions sources.**

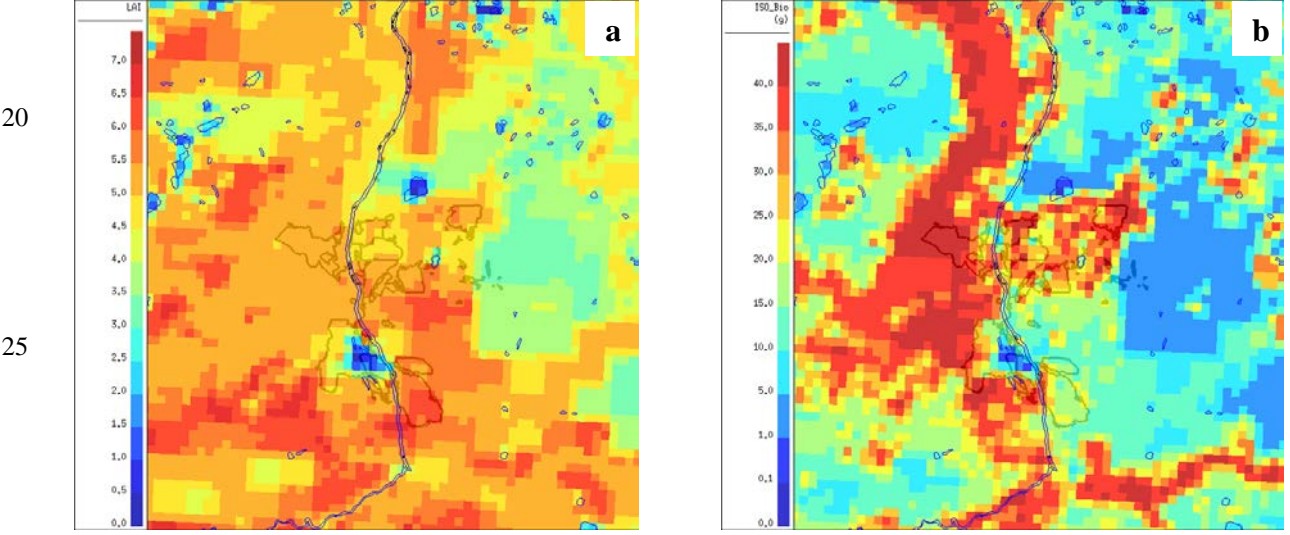

30  **Figure 5:  (a) Leaf Area Index and (b) peak summer isoprene emissions computed on the 2.5-km for a portion of the 2.5-km OS grid centred on the AOSR study area from the original BELD3 database.  The gray lines indicate the cleared areas within the boundaries of the six AOSR mining and processing facilities (cf. Figure 1).**





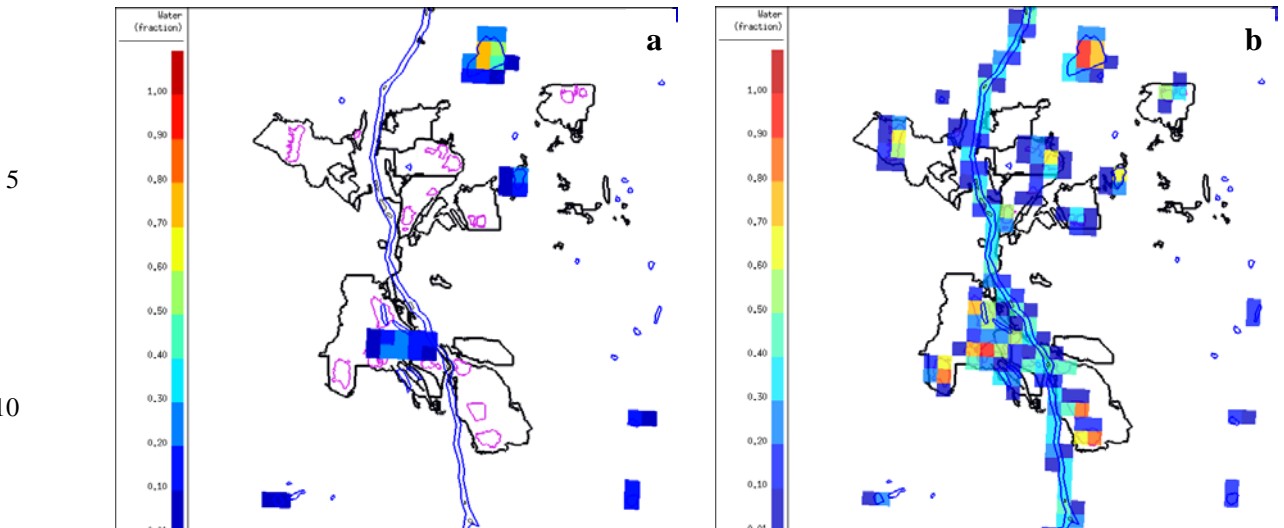

**Figure 6.  (a) Inland water coverage on the 2.5-km for a portion of the 2.5-km OS grid centred on the six AOSR mining and processing facilities generated from the original land cover database (only natural  lakes); and (b) modified inland water coverage including tailings ponds and rivers. The black  and pink lines in panel (a) indicate the cleared-land areas and the tailings ponds within the boundaries of the six AOSR mining and processing facilities, whereas the blues lines in panel (a) mark the boundaries of natural lakes and rivers.**

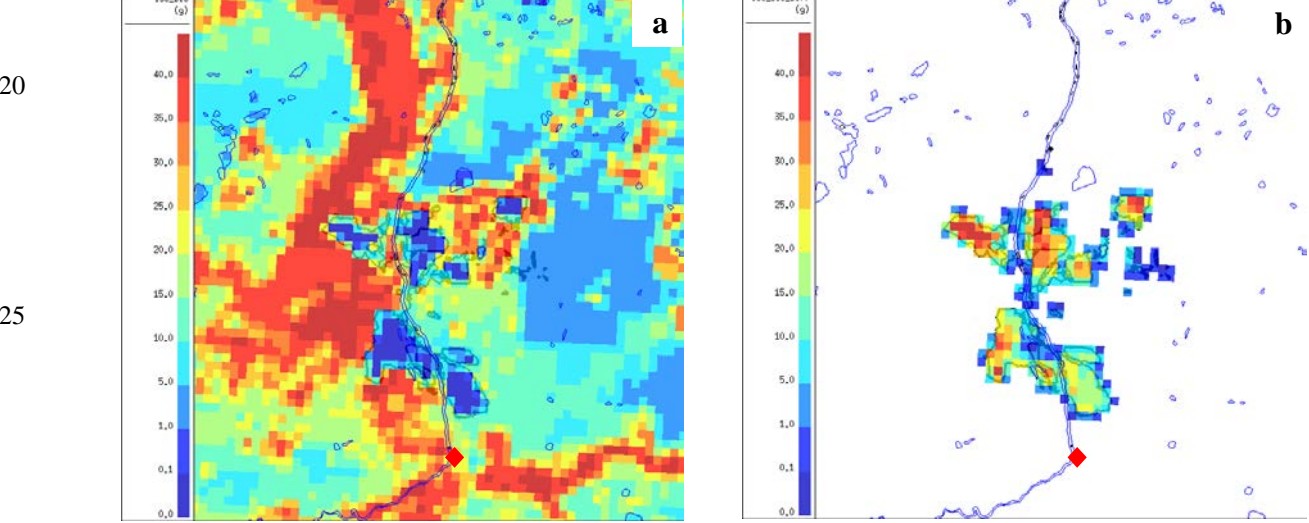

**Figure 7:  (a) Modified biogenic isoprene emissions for a portion of the 2.5-km OS grid centred on the AOSR study area and (b) difference between the original and the modified isoprene emissions (original – modified).  The gray lines indicate the cleared-land areas within the boundaries of the OS mining facilities.  The location of Fort McMurray is indicated by the diamond symbol.**



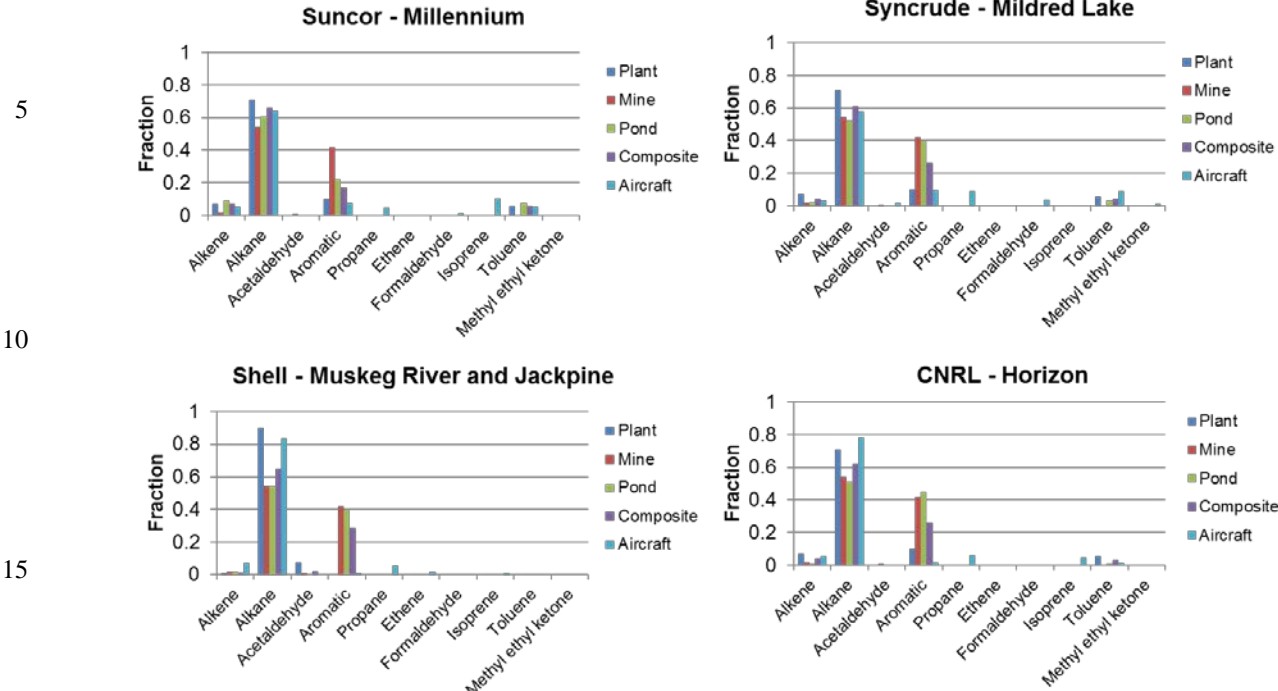

**Figure 8: Comparisons of facility-specific VOC speciation profiles for ADOM-2-mechanism for four AOSR mining**
20 **facilities used for the base-case study with facility-specific profiles derived from aircraft observations. Different VOC**
**speciation profiles for plants, mine faces, and tailings ponds were used for the base-case study. The "composite"**
**VOC speciation profile for the base case is an emissions-weighted combination of the plant, mine-face, and tailings-**
**pond profiles for each facility to allow comparison with the aircraft-based facility-specific VOC speciation profiles.**





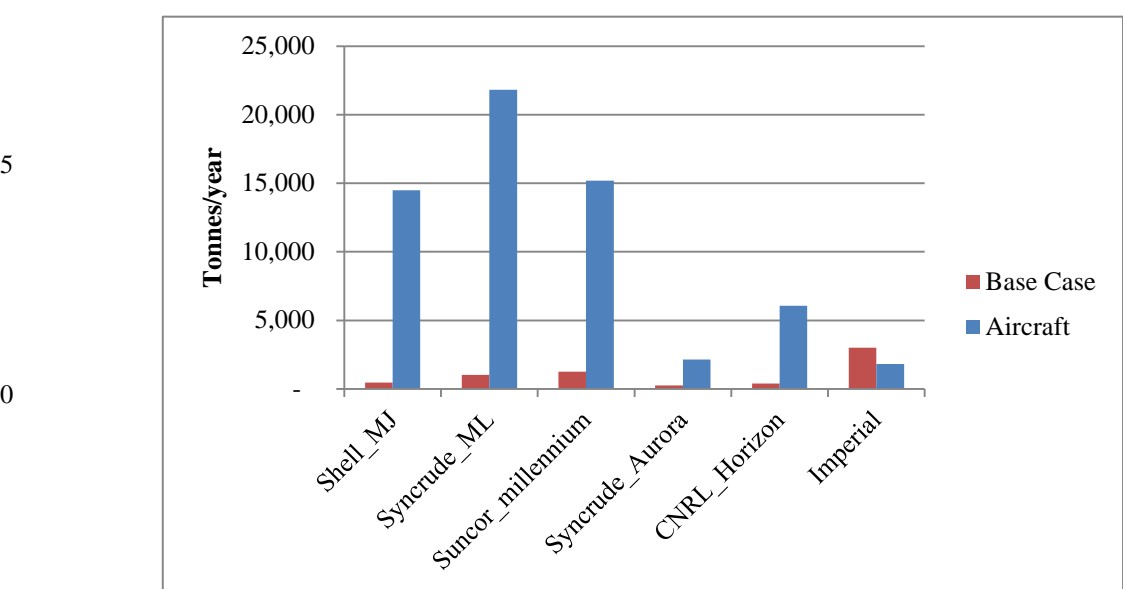

**Figure 9: Comparison of annual PM$_{2.5}$ emissions between base-case and aircraft-observation-based estimates for the six AOSR mining facilities.**

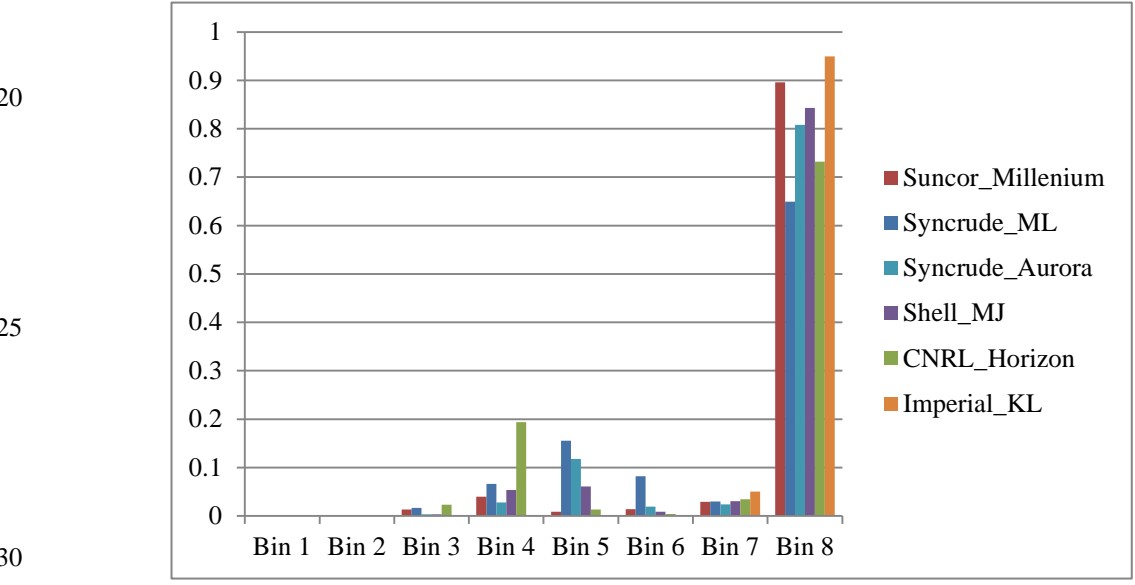

**Figure 10: PM$_{2.5}$ size distribution derived from the aircraft observations for the six AOSR mining facilities.**



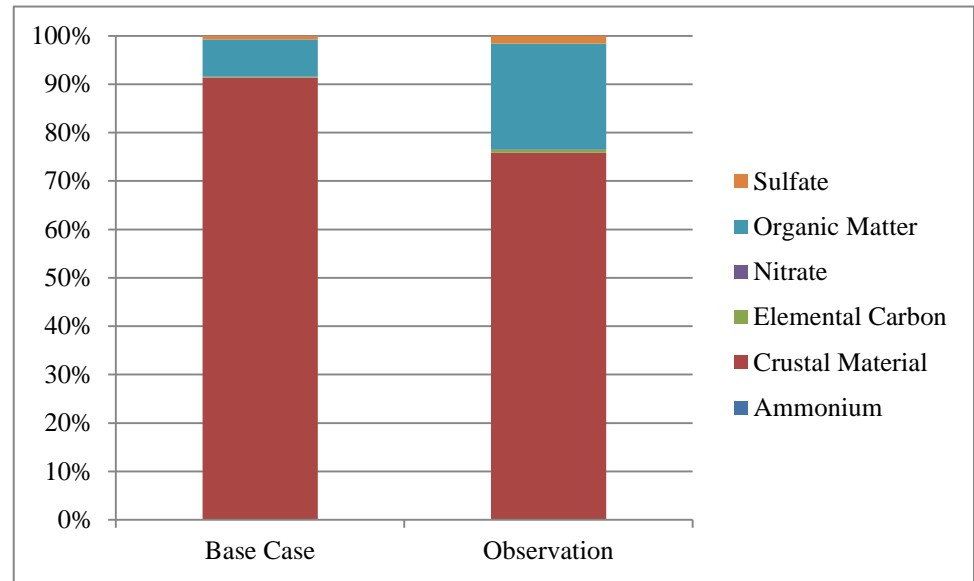

10  **Figure 11:  Comparison of the fugitive-dust PM speciation profile used for the base-case study and the one compiled from soil analyses from Wang et al. (2015) for the AOSR mining facilities.**

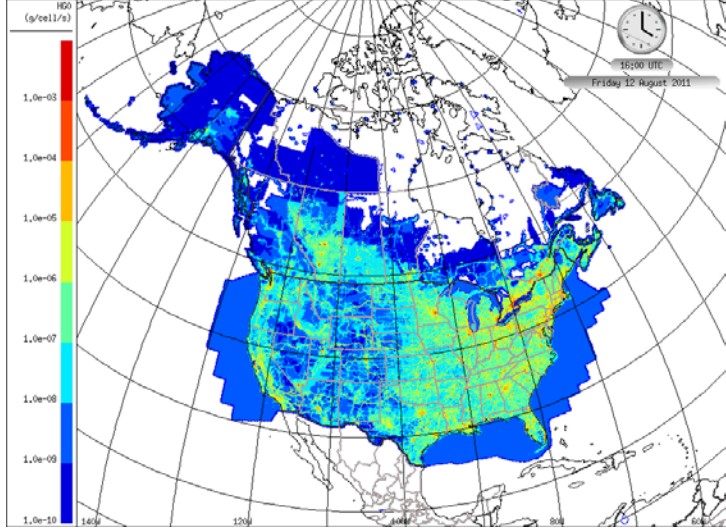

25  **Figure 12:  Spatial distribution of Phase 3 elemental mercury emissions for Canada and the U.S. for the 10-km continental model grid for one afternoon hour in August.  Note logarithmic spacing of the emissions contour intervals; white areas have emissions less than 10⁻¹⁰ g/cell/s.**