# Peer review of "Emissions Preparation and Analysis for Multiscale Air Quality Modelling over the Athabasca Oil Sands Region of Alberta, Canada"

_Atmospheric Chemistry and Physics, 2017_

## Referee Comment (RC1) · Anonymous Referee #2 · 13 Mar 2018

In this manuscript, the authors compiled an emission inventory specific to the oil sands region of Alberta for further use in an air quality model. They harmonized several emission inventories and updated with additional data sources specific to the modeling year (2013). They describe in details how they accomplished this goal.

General comments on Zhang et al.

As it stands, this paper is largely a straightforward emission development application and its format resembles a report more than a scientific paper. The abstract is very lengthy similar to a report executive summary. Same information is provided repetitively. The paper needs to be a lot more concise especially when the same information

is available elsewhere (references). If qualitative arguments like "significantly impacted" are used to characterize the results, they need to be supported by facts and figures. While subject to a variety of weaknesses and not developing any new approaches or containing extremely unique analyses, it is a useful paper and fits well in this special issue. Essentially, it serves as a basis for other papers in the same special issue. The paper could be improved by presenting a linkage to uncover what lessons are learned from the air quality applications that have used this inventory.

Specific Comments on Zhang et al. Abstract - The current form is too long and is read like a 2-page executive report summary rather than a research abstract. Focus on key messages and make the abstract more concise.

Page 4 – The summary on the first two phases is repetitive. Much of the text in the first paragraph can be excluded. Page 6-8 already give more than sufficient background information (which can still be further condensed in my opinion).

Page 5 (from Line 14-25) – This section is not needed.

Page 6 (Line 28) If a setup for processing 2006 APEI already exists, it is not clear why 2010 APEI could not be formatted to leverage that. Information in the APEI is simple, e.g., province, SCC, pollutant, emissions.

Page 7 (Line 12) Why didn't each facility level scale up by facility-specific scaling factor as supposed to a uniform constant of 2.6? For example, VOC emissions from the CNRL facility should have scaled up by about a factor of 10. Would not this offer better spatial representation? Also, were there particular VOC sources missing in the CEMA inventory? The VOC discrepancy is surprising given that the CEMA inventory is a bottom-up inventory. I suspect that the 2010 NPRI values were too high considering much lower estimates in the 2013 NPRIv2. This point should be clarified because this updated inventory relied heavily on both CEMA and NPRI. The referenced report does not address this point.

[Figure]

Page 8 (Line 1) Treating industrial non-stack sources as area sources (e.g., distributing within the facility boundary and specific to source type) is appropriate. However, this is quite a standard approach, NOT a new approach as claimed.

Page 10 (first paragraph) The US EPA's trend suggests a reduction of NOx by 8% and SO2 by 23% between 2011 and 2013 [https://www.epa.gov/air-emissions-inventories/air-pollutant-emissions-trends-data]. This should be mentioned in the paper.

Page 10 (Line 20) This comment is related to my comment made above on the 2010 NPRI inventory. I find the difference between version 1 and 2 of the 2013 NPRI to be disturbing. What is the level of confidence when using NPRI inventory? Would you recommend extra procedures to ensure that the NPRI values are representative?

Page 11 (Line 16) The argument is that VOC emissions in the CEMA inventory are underestimated with no reasons given. What are your justifications of using the CEMA inventory to represent source profile (to allocate fugitive VOC emissions among mine faces, tailing ponds, etc.)? If the underlying assumptions of VOC estimates are vastly different between NPRI and CEMA, this approach would not be appropriate especially if VOC speciation profiles differ significantly among these sources.

Page 11 (Line 26) How was the 2011 UOG emissions projected to 2013? Page 12 (Line 5) From what is shown in Figure S2, the spatial distribution of UOG emissions does not change very much with a few exceptions. Perhaps, include UOG emission spatial plots to support your statement.

Page 12 (Line 20) SMOKE reference has been mentioned – no need to repeat here. Page 13 (Line 14) I agree that the temporal profiles for fugitive VOC are dependent of temperature, but that's not the only factor. Wind speed (and snow cover) can also affect emission flux. How did you address wind speed effects?

Page 14 (Line 29 and also Page 15 Line 5) Need a reference to those tests that show

'significant (as the authors claimed)' effects of the artificial water bodies. Be specific about what effects are.

Page 15 (2nd paragraph) The authors' effort in updating landuse/land cover is to be commended. This is rarely done in air quality applications. Since BELD4 is already available, would future work require this step?

Page 15 (Line 9) Instead of using qualitative description such as 'significantly impacted', present facts. , e.g. change in biogenic VOCs (%).

Page 16 (Line 25; Figure S5, S6) Why not comparing 'annual' CEMS to the NPRI inventory? As the authors noted, there could be large variation in the CEM measurements. Using two-month worth of data does not give me confidence in the comparison shown in Figure S5 and S6. Alternatively, you can report NPRI emissions that are temporally allocated to the same two months.

Page 17 (Line 14) How are the aircraft-estimated VOC emissions spatially allocated within each facility for modeling? The numbers shown in Table 6 are in tonnes/year, but aircraft observations are only available during the field campaign. Hourly aircraft data was scaled to daily rate based on temperature profile during the flight day (Li et al). Were the aircraft-derived estimated annualized by scaling daily values with temperature (as was done in the base case)?

Page 18 (Line 1) The observation-based VOC profiles show lower higher-aromatic fractions. Could this be a result of flight altitude (150-1470 m) which could be more influenced by stack emissions rather than surface sources like tailing ponds?

Page 18 (Line 11) My understanding is that the aircraft-derived emissions are at facility level so any scaling would be the same across all grid cells within individual facilities. Why are the ratios shown in Figure S7 varying within individual facilities?

Page 19 (Line 29) I understand that the aircraft-derived emissions are only used for summertime modeling. The emission comparison needs to be limited to just summer

months (so summing basecase emissions for these months), instead of assuming constant daily emissions throughout the year to get annualized value. Reporting 61,500 tpy of PM2.5 is totally misleading and overemphasizing the role of wind-blown dust.

Page 20 (Line 29) It is interesting to see a high fraction of OC from soil in this area. Is there any literature that reports similar finding?

Summary Section – Information on Phase I and Phase II is repetitive (page 23 and 24). The authors should focus on Phase III and some new insights that this work has revealed. Some recommendations made in this paper are reasonable. Since the basecase and sensitivity emissions were already applied in GEM-MACH, what are the lessons learned? On page 27 (last paragraph), the authors suggest that fugitive dust emissions are underestimated based on air quality modeling (by how much?). The authors should distill some scientific findings to share with readers for other pollutants (such as NOx, VOC, or SO2) based on other air quality modeling activities referred in the paper.

Figure 1. This figure needs improvement on print quality/resolution. Table S3. Why was the 2010 CEMA off-road inventory selected? Did you compare these emissions with other inventories? Figure S3. Resolution issue. Color scales are not readable. And why is ethane shown instead of total VOC?

---

## Referee Comment (RC2) · Anonymous Referee #1 · 17 Apr 2018

The authors describe the anthropogenic and biogenic emissions datasets developed for the global air quality model GEM-MATCH to simulate air quality (AQ) in summer 2013 over the Athabasca Oil Sands Region (AOSR) of Canada. The paper provides a detailed description of the number of datasets and emission inventories that are used to generate a new hybrid emissions inventory for high-resolution AQ modeling over the AOSR. I recommend the manuscript for final publication in ACP after addressing these questions and comments:

The text needs to be shortened, e.g. some parts (Abstract, Summary) are too long and repetitive.

[Figure]

The paper uses too many acronyms that are hard to follow. I suggest adding a table to introduce all the acronyms that are used in the paper.

One of key improvements of the new emission dataset is the improved biogenic VOC emissions due to the use of the new land use map (forest clearing and water and ponds). The regional and global AQ models typically use the outdated static vegetation and LAI maps, hence introducing large uncertainties to the biogenic VOC simulations. Was the land-use map modified to improve the meteorological simulations as well?

The paper reports that the aircraft did measure high isoprene from the Suncor Millenium/Steepbank and the CNRL Horizon facilities. What are the sources emitting the high amount of isoprene?

The paper mentions Stroud et al., 2017 study to model SOA over the AOSR by using the emissions inventory developed in this study. There a number of uncertainties in the emission inventories that affect the modeled SOA levels. First, does this emission dataset include intermediate VOCs (IVOCs) emissions from the anthropogenic sources in the AOSR? How various long-chain alkane and other species are lumped in the developed inventory, which can affect the SOA production in the model? Did the emission dataset characterize the semi-volatile organic species (SVOCs)? This also depends on the volatility distribution of the primary OA emissions. Not sure if the POA is assumed to be non-volatile in this dataset. Are the improvements for such SOA precursors (S- and I-VOCs) in this new emission development over the existing inventories used for the regulatory purposes?

The new emission dataset also includes some emission estimates based on the aircraft measurements and mass balance approach. I think the authors need to put more emphasis on the use of the top-down emission estimates in the paper. As the Summary section discusses, there are some uncertainties associated with the top-down emission datasets. However, in the text it isn't clear the distinction between the top-down and bottom-up emission datasets and their use in the AQ models.

There were studies in the US to improve the emission inventories for the oil and gas sector and simulate air quality by taking advantage of the top-down emission estimates for NOx, CH4 and VOCs from the oil and gas sector. Unfortunately, the findings of those studies aren't discussed in this paper. Below are some references:

1) Karion, A.; Sweeney, C.; Petron, G.; Frost, G.; Hardesty, R. M.; Kofler, J.; Miller, B. R.; Newberger, T.; Wolter, S.; Banta, R.; Brewer, A.; Dlugokencky, E.; Lang, P.; Montzka, S. A.; Schnell, R.; Tans, P.; Trainer, M.; Zamora, R.; Conley, S., Methane emissions estimate from airborne measurements over a western United States natural gas field. Geophys. Res. Lett. 2013, 40, (16), 4393-4397.

2) Peischl, J.; Ryerson, T. B.; Aikin, K. C.; de Gouw, J. A.; Gilman, J. B.; Holloway, J. S.; Lerner, B. M.; Nadkarni, R.; Neuman, J. A.; Nowak, J. B.; Trainer, M.; Warneke, C.; Parrish, D. D., Quantifying atmospheric methane emissions from the Haynesville, Fayetteville, and northeastern Marcellus shale gas production regions. J. Geophys. Res.-Atmos. 2015, 120, (5), 2119-2139.

3) Ahmadov, R.; McKeen, S.; Trainer, M.; Banta, R.; Brewer, A.; Brown, S.; Edwards, P. M.; de Gouw, J. A.; Frost, G. J.; Gilman, J.; Helmig, D.; Johnson, B.; Karion, A.; Koss, A.; Langford, A.; Lerner, B.; Olson, J.; Oltmans, S.; Peischl, J.; Petron, G.; Pichugina, Y.; Roberts, J. M.; Ryerson, T.; Schnell, R.; Senff, C.; Sweeney, C.; Thompson, C.; Veres, P. R.; Warneke, C.; Wild, R.; Williams, E. J.; Yuan, B.; Zamora, R., Understanding high wintertime ozone pollution events in an oil- and natural gas-producing region of the western US. Atmos. Chem. Phys. 2015, 15, (1), 411-429.

4) Gilman, J. B.; Lerner, B. M.; Kuster, W. C.; de Gouw, J. A., Source Signature of Volatile Organic Compounds from Oil and Natural Gas Operations in Northeastern Colorado. Environ. Sci. Technol. 2013, 47, (3), 1297-1305.

---

## Author Comment (AC1) · 29 May 2018

General responses from the authors:

We greatly appreciate the time and effort spent by both reviewers in providing their helpful comments. One comment shared by both reviewers is that some parts of the text need to be shortened. We have taken this comment seriously and have shortened the paper accordingly. We are also very grateful for other general and specific comments from the reviewers, and we believe that by addressing these comments the manuscript has been substantially improved. In addition, revisions were also made to address comments received externally from Canada's Oil Sands Innovation Alliance (COSIA) and internally from two other divisions within our organization (Environment and Climate Change Canada), who are responsible for compiling Canada's national emissions inventories of criteria air contaminants and greenhouse gases.

Below are our point-by-point responses (in blue font) to all of the review comments:

**Anonymous Referee #1**:

General comments:

The authors describe the anthropogenic and biogenic emissions datasets developed for the global air quality model GEM-MATCH to simulate air quality (AQ) in summer 2013 over the Athabasca Oil Sands Region (AOSR) of Canada. The paper provides a detailed description of the number of datasets and emission inventories that are used to generate a new hybrid emissions inventory for high-resolution AQ modeling over the AOSR. I recommend the manuscript for final publication in ACP after addressing these questions and comments

We appreciate your comments very much. See below for our responses to your specific comments.

Specific Comments:

1) The text needs to be shortened, e.g. some parts (Abstract, Summary) are too long and repetitive.

   Following your comment and the other reviewer's comments, we have shortened the Abstract and the Summary and Future Work sections, moved Sections 2.1.1 and 2.1.2 to the Supplemental Material, and revised some other sections of the paper.

2) The paper uses too many acronyms that are hard to follow. I suggest adding a table to introduce all the acronyms that are used in the paper.

   When we were preparing this manuscript, we were also concerned about this problem and therefore we included a table (Table S1) in the Supplemental Material to define all of the acronyms. We have now moved that table to the main paper as an Appendix to make it more accessible to the reader.

3) One of key improvements of the new emission dataset is the improved biogenic VOC emissions due to the use of the new land use map (forest clearing and water and ponds). The regional and global AQ models typically use the outdated static vegetation and LAI maps, hence introducing large uncertainties to the biogenic VOC simulations. Was the land-use map modified to improve the meteorological simulations as well?

   Yes, we also modified the land-use map to improve the meteorological simulations. To address this point we have added one more plot in the Supplemental Material (Figure S3) to show the impact of the modified land use on model-predicted PBL height. The impact can be a factor of two or more.

4) The paper reports that the aircraft did measure high isoprene from the Suncor Millenium/ Steepbank and the CNRL Horizon facilities. What are the sources emitting the high amount of isoprene?

   Based on aircraft observations and preliminary lab test results, the isoprene seems to come from bitumen vapor emissions. We are curious about the isoprene source as well and are conducting further field studies this year (2018). We have made two text additions: "likely originated from bitumen vapor emissions from" and "Further studies are needed to confirm the source of non-biogenic isoprene emissions" to Page 15, Lines 22 and 24-25 (note that page and line numbers have changed after the revisions to the paper).

5) The paper mentions Stroud et al., 2017 study to model SOA over the AOSR by using the emissions inventory developed in this study. There a number of uncertainties in the emission inventories that affect the modeled SOA levels. First, does this emission dataset include intermediate VOCs (IVOCs) emissions from the anthropogenic sources in the AOSR? How various long-chain alkane and other species are lumped in the developed inventory, which can affect the SOA production in the model? Did the emission dataset characterize the semi-volatile organic species (SVOCs)? This also depends on the volatility distribution of the primary OA emissions. Not sure if the POA is assumed to be non-volatile in this dataset. Are the improvements for such SOA precursors (S- and I-VOCs) in this new emission development over the existing inventories used for the regulatory purposes?

   IVOC and SVOC species were not measured by the aircraft. Canister samples of VOCs up to C12 and the aerosol mass spectrometer (AMS) $PM_1$ organic aerosol were measured. The box flights encircling individual facilities were able to calculate net fluxes of VOCs and $PM_1$ organic aerosol from entire facilities based on the measured concentrations and wind speeds. These measurement-derived VOC emissions were used as input to the GEM-MACH sensitivity simulation performed by Stroud et al. and lumped to a long-chain alkane (ALKA), two aromatic species (AROM, TOLU), a long-chain alkene (ALKE) and non-volatile POA. In GEM-MACH, the long-chain alkane has an SOA yield representative of a C10 species, but only a small fraction of the ALKA lumped species is assumed to be C10 or greater, so GEM-

MACH does not account for the IVOC SOA formation by lumping the IVOCs into the ALKA species.

The aircraft observations suggest POA emissions and/or rapid SOA formation over and just downwind of the open pit mines.  It is not clear the relative proportion of each. The O/C ratio of the organic aerosol over or just downwind of the open pit mines suggests it is already partially aged, as a ratio greater than 0.4 was already observed. This favours the rapid SOA formation mechanism from reactive precursors such as anthropogenic VOCs and/or from the SVOCs that only need one oxidation step to reach an O/C ratio of 0.4 (based on a box model study).  The IVOCs likely need more time than one hour after emission to start dominating the SOA production.  The previous box model study, by Liggio et al. (2015), was initialized at the first transect of a downwind Lagrangian flight pattern and a large concentration of organic aerosol was already measured at this point. The box modelling suggested that high concentrations in the IVOC range were needed thereafter to account for the near-steady organic aerosol concentration, where continued SOA production was needed to balance dilution loss in the plume.

It is likely that the model emissions are missing the IVOCs and that the aircraft-measured organic aerosol already contains some SOA from the reactive anthropogenic VOC precursors and from the SVOCs emitted that only need one oxidation step to form low-volatile material. This rapid SOA mass production would be, at least partially, accounted for in the new model POA emissions measured by aircraft box method. The current GEM-MACH version does not include the volatility basis set (VBS) approach to account for POA volatility, but rather assumes the POA to be non-volatile.  A research version of GEM-MACH has recently been coded with the VBS approach; however, the IVOC emission factors remain highly uncertain as does the IVOC/SVOC aging scheme.  A new aircraft field study in the Oil Sands region is planned for this year (2018) that will measure IVOC gaseous species over the open pit mines, as well as employ an electrospray-ionization AMS to measure organic aerosol composition. This will provide additional information to better address these uncertainties.

6) The new emission dataset also includes some emission estimates based on the aircraft measurements and mass balance approach. I think the authors need to put more emphasis on the use of the top-down emission estimates in the paper. As the Summary section discusses, there are some uncertainties associated with the top-down emission datasets.  However, in the text it isn't clear the distinction between the top-down and bottom-up emission datasets and their use in the AQ models.

This is a very good point.  We have added the modifier "top-down" to the titles of Sections 4.2 and 4.3 and revised the contents of these sections to differentiate between the top-down and bottom-up emissions estimates and to emphasize the top-down emissions.

7) There were studies in the US to improve the emission inventories for the oil and gas sector and simulate air quality by taking advantage of the top-down emission estimates for NOx,

CH4 and VOCs from the oil and gas sector. Unfortunately, the findings of those studies aren't discussed in this paper. Below are some references:

Thank you very much for pointing out these studies in the U.S. We found three of them to be very relevant to our study and we have cited them in Sections 4.2 and 4.3 when aircraft-measurement-based VOC and PM emissions estimates are discussed.

**Anonymous Referee #2**:

General comments:

As it stands, this paper is largely a straightforward emission development application and its format resembles a report more than a scientific paper. The abstract is very lengthy similar to a report executive summary. Same information is provided repetitively. The paper needs to be a lot more concise especially when the same information is available elsewhere (references). If qualitative arguments like "significantly impacted" are used to characterize the results, they need to be supported by facts and figures. While subject to a variety of weaknesses and not developing any new approaches or containing extremely unique analyses, it is a useful paper and fits well in this special issue. Essentially, it serves as a basis for other papers in the same special issue. The paper could be improved by presenting a linkage to uncover what lessons are learned from the air quality applications that have used this inventory.

Thank you very much for these comments. As you point out, this emissions paper serves as a foundation for several of the modelling papers in this special issue and hence it makes a useful contribution to this special issue. Following your specific comments, we have shortened the Abstract and the Summary and Future Work sections, moved Sections 2.1.1 and 2.1.2 to the Supplemental Material, and revised some other sections of the paper. We have replaced some descriptions like "significantly impacted" by more fact-based statements. We also linked this paper more closely with the air quality modelling papers in the special issue to summarize what emissions-based lessons were learned from this research activity. More details are provided in the responses to your specific comments.

Specific Comments:

1) Abstract - The current form is too long and is read like a 2-page executive report summary rather than a research abstract. Focus on key messages and make the abstract more concise.

   We have shortened the abstract by more than one-third.

2) Page 4 – The summary on the first two phases is repetitive. Much of the text in the first paragraph can be excluded. Page 6-8 already give more than sufficient background information (which can still be further condensed in my opinion).

We agree that much of the text in the first paragraph is repetitive. We have condensed this paragraph (from 407 words to 280 words). We have also moved most part of Section 2.1 ("Review of emissions inventories used for JOSM Phases 1 and 2 AQ modelling", Pages 6-8, now Page 5) and two corresponding tables (Tables 1 and 3) to the Supplemental Material.

3) Page 5 (from Line 14-25) – This section is not needed.

This section briefly describes the following sections of the paper. As a common practice, it serves as a transitional paragraph from the Introduction section to the main body of the paper. We did not remove this section completely, but we did shorten it considerably.

4) Page 6 (Line 28) If a setup for processing 2006 APEI already exists, it is not clear why 2010 APEI could not be formatted to leverage that. Information in the APEI is simple, e.g., province, SCC, pollutant, emissions.

Canada's Air Pollutant Emission Inventory (APEI) is a comprehensive anthropogenic emissions inventory that is prepared to fulfill various national and international reporting obligations and to provide data for air quality modelling. However, additional preparation steps are needed to transfer the emissions database underlying the published APEI data to a format that is suitable for processing emissions for AQ modelling, such as a further breakdown to more detailed source types for SCC assignment. More detailed emissions data are also needed than are stored in the annual APEI, such as (a) monthly on-road and off-road emissions and (b) process-level reporting such as on-road evaporative emissions vs. exhaust emissions. Other countries have similar gaps between their national reported inventories and their emissions-processing-ready inventories. For example, the U.S. EPA compiles emissions for their policy modelling platform (https://www.epa.gov/air-emissions-modeling/emissions-modeling-platforms) based on their NEI (National Emissions Inventory, https://www.epa.gov/air-emissions-inventories/national-emissions-inventory-nei). We have added the following explanation to this line: ", which requires the published APEI data to be transferred to a format that is suitable for processing emissions for AQ modelling as well as the addition of more detailed emissions data, such as monthly on-road and off-road emissions and process-based separation of emissions from some sectors (e.g., evaporative vs. exhaust emissions from on-road vehicles)" (now in Section S1 of the Supplemental Material, Page 1, Lines 22-25).

5) Page 7 (Line 12) Why didn't each facility level scale up by facility-specific scaling factor as supposed to a uniform constant of 2.6? For example, VOC emissions from the CNRL facility should have scaled up by about a factor of 10. Would not this offer better spatial representation? Also, were there particular VOC sources missing in the CEMA inventory? The VOC discrepancy is surprising given that the CEMA inventory is a bottom-up inventory. I suspect that the 2010 NPRI values were too high considering much lower estimates in the 2013 NPRIv2. This point should be clarified because this updated inventory relied heavily on both CEMA and NPRI. The referenced report does not address this point.

This is a good question.  First of all, we do not think that the CEMA inventory misses any significant sources of VOC emissions, but it is a static, "one-off" inventory.  For the ongoing NPRI inventory, whose preparation is a requirement under the authority of the Canadian Environmental Protection Act, owners or operators of facilities that meet published reporting requirements are required to report to the NPRI annually.  However, various estimation methods with different uncertainties may be used. (See https://www.canada.ca/en/environment-climate-change/services/national-pollutant-release-inventory/publications/guide/exemptions-exclusions.html#a3_8 for the list of methods that the facility can use to estimate emissions.)  For the same activity data, different estimation methods can result in different emission estimates for a bottom-up inventory.  Facilities are also allowed to update their emissions after the official NPRI inventory is published if they deem it necessary due to the availability of new information or new estimation methodologies.  As a result, the emissions reported by a facility for a given year can vary from inventory version to inventory version.  In fact, as you commented below, you found the "difference between version 1 and 2 of the 2013 NPRI to be disturbing", but we view this aspect of the NPRI to be positive since it allows better emissions estimates to be included after the initial submission.

It is therefore difficult to choose the most suitable inventory.  Before Phase 1, our understanding of the emissions from the Oil Sands facilities was limited.  Our decision in Phase 1 to scale the CEMA VOC emissions to the 2010 NPRI level for the OS facilities using a uniform factor was based mainly on three considerations: (1) emissions in the NPRI inventory were required by law to be reported by each facility, and we assumed that each facility is best placed to know its own emissions; (2) the CEMA inventory had the lowest total VOC emissions for these five facilities compared to four other inventories (ECCC & AEP, 2016); and (3) large uncertainties may be present in the reported NPRI inventory (and also in the CEMA inventory) and the use of a uniform scaling factor will not affect the impact of VOC emissions from the OS facilities as a whole.  We had already explained the first and second reasons in the paper and we have now added the following text to clarify the third reason: "as large uncertainties may exist in both inventories and the use of a uniform scaling factor should not affect the impact of VOC emissions from the OS facilities as a whole" (now in Section S1 of the Supplemental Material, Page 2, Lines 7-8).  Note that after we gained more knowledge of the emissions from this region through Phases 1 and 2 of the study, we treated VOC emissions separately for each facility in Phase 3.

6) Page 8 (Line 1) Treating industrial non-stack sources as area sources (e.g., distributing within the facility boundary and specific to source type) is appropriate. However, this is quite a standard approach, NOT a new approach as claimed.

To the contrary, our understanding is that treating non-stack sources in an individual industrial facility as area sources is not a standard approach. Such emissions are usually allocated to a single grid cell based on the single pair of latitude–longitude values provided for the non-stack sources of that facility, which is clearly not optimal when a facility is very large and spans multiple grid cells.  Moreover, it is also not a common practice to develop a

set of facility-specific, process-specific spatial surrogate fields for each facility to allocate non-stack source emissions.  We believe our approach to be unique; at least we are not aware of any other groups that have applied a similar approach, which may not be that surprising given the unusually large spatial scale of the oil sands facilities considered in this study and the fine horizontal grid spacing (2.5 km) that was used by the model.

7) Page 10 (first paragraph) The US EPA's trend suggests a reduction of NOx by 8% and SO2 by 23% between 2011 and 2013 [https://www.epa.gov/air-emissionsinventories/air-pollutant-emissions-trends-data]. This should be mentioned in the paper.

Thank you very much for pointing out this. We have added the following sentence to the end of this paragraph:
"Note, however, that the U.S. EPA's emissions trend data set suggests a reduction of $NO_x$ emissions by 8% and $SO_2$ emissions by 23% between 2011 and 2013 (https://www.epa.gov/air-emissionsinventories/air-pollutant-emissions-trends-data)" (now Page 6, Lines 24-26).

8) Page 10 (Line 20) This comment is related to my comment made above on the 2010 NPRI inventory. I find the difference between version 1 and 2 of the 2013 NPRI to be disturbing. What is the level of confidence when using NPRI inventory? Would you recommend extra procedures to ensure that the NPRI values are representative?

As we mentioned in the paper and explained above, NPRI is the Canada's legislated inventory of large point sources and is based on emissions reported by facilities.  Various methods with different inherent uncertainties can be used for estimation of emissions.  We are confident with emissions derived from CEMS measurement, such as $SO_2$ and $NO_x$ emissions, although even for that method, large uncertainty still remains when a facility experiences upset conditions and the CEMS instruments are bypassed.  Emissions derived from Engineering Judgement or general Emission Factors, for example, are less reliable.  We must trust that facilities have done their best to report their emissions, but we must also always keep in mind that the emissions reported may not be representative.  Other inventories, such as the U.S. NEI, have similar uncertainties.  This is one of the reasons for the 2013 OS field study: to allow independent emissions estimates to be produced based on aircraft observations, thus facilitating evaluation of reported NPRI facility emissions.  However, such field studies are very difficult and expensive to undertake and are only possible under special circumstances for a very small number of facilities.

9) Page 11 (Line 16) The argument is that VOC emissions in the CEMA inventory are underestimated with no reasons given. What are your justifications of using the CEMA inventory to represent source profile (to allocate fugitive VOC emissions among mine faces, tailing ponds, etc.)? If the underlying assumptions of VOC estimates are vastly different between NPRI and CEMA, this approach would not be appropriate especially if VOC speciation profiles differ significantly among these sources.

We explained in Section 2.1.1 that the VOC emissions reported in the CEMA inventory might be underestimated (Page 7, Line 7-11, now in Supplemental Material, Section S1, Page 2, Lines 1-5). The relevant text reads:

"The 2010 NPRI was also used to scale the CEMA facility-total VOC emissions for the five AOSR surface mines active at that time (Figure 1), since it was found that the CEMA inventory had the lowest total VOC emissions for these five facilities compared to four other inventories (ECCC & AEP, 2016) and the NPRI is Canada's legislated inventory of large point sources based on emissions reported by facilities."

As mentioned in this same paragraph, fugitive VOC emissions are reported to NPRI as facility-total emissions without differentiation between source type (i.e., mine faces, tailings ponds, and extraction/upgrading plants), whereas VOC area-source emissions in the CEMA inventory are reported by process, including fugitive VOC emissions from tailings ponds, plants, and mine faces.  We therefore used split factors from the CEMA inventory to separate NPRI VOC emissions by source type because there is no better information than the CEMA inventory available to estimate the relative contributions of sub-facility, process-level emissions.  We explained this reasoning in the paper as well for Phase 3 (now Page 8, Lines 1-6), where the text reads:

"Surface-level fugitive VOC emissions are reported to NPRI as facility-total emissions without differentiation between source type (i.e., mine faces, tailings ponds, and extraction/upgrading plants).  To distribute 2013 NPRI fugitive VOC emissions spatially within an OS mining facility, process allocation factors calculated from the process-specific fugitive VOC emissions in the 2009/10 CEMA inventory for each AOSR mining facility were used to allocate fugitive VOC emissions between mine faces, tailings ponds, and plants (similar to the procedure used in Phase 2; see ECCC & AEP, 2016)."

Aircraft measurements conducted during the 2013 field study are not helpful here because they can only be used to calculate emissions estimates at the facility level.  We understand this limitation and we recommended this as an area for improvement in the Summary and Future Work section (Page 23, Lines 21-28).  We have modified the text further for clarity. The text now reads (new text is shown in black font):
"Moreover, these aircraft measurements were carried out at the facility level, but within these very large facilities the individual VOC species emitted from mine faces, tailings ponds, and plants can be very different.  More aircraft measurements, especially at other times of year, and additional measurements of emissions at the sub-facility level, from mine faces, tailings ponds, and plants for multiple AOSR facilities are needed to confirm and augment the findings of the 2013 field study and to further improve emissions factors, temporal profiles, and chemical speciation profiles used for OS emissions inventories and emissions processing(e.g., Small et al., 2015; Stantec Consulting Ltd. et al., 2016)."

10) Page 11 (Line 26) How was the 2011 UOG emissions projected to 2013? Page 12 (Line 5) From what is shown in Figure S2, the spatial distribution of UOG emissions does not change very much with a few exceptions. Perhaps, include UOG emission spatial plots to support your statement.

The projection from 2011 UOG emissions to 2013 is based on activity data and a methodology described in a Clearstone Engineering Ltd. letter report entitled "Documentation for UOG emission inventory extrapolation database (ExtrapolateR.accdb)". We have changed the sentence to read (new text shown in black font): "This new subinventory was then projected by ECCC to 2013 for inclusion in the 2013 APEI based on activity data and a methodology described in a letter report from Clearstone Engineering Ltd. (2014d)." (now Page 8, Lines 17-19). We have also added this report to the References section.

The majority of the facilities in the UOG inventory are conventional oil & gas wells and batteries, which are not abundant in the immediate vicinity of the AOSR. That is the reason why the spatial distribution of UOG emissions does not change very much in the original Figure S2. As suggested, we prepared some new UOG emissions spatial plots to support our statement that "some UOG facilities that existed in 2000 have been closed while many new facilities have opened since 2000". However, it is difficult to see in these new plots whether the differences are due to changes in the numbers and locations of facilities or to changes of emissions magnitude. We then plotted the spatial distribution of UOG facilities over a larger area near the AOSR as shown below. We believe these new plots, which are now both superpositions of the two UOG versions but in different order, support our statement well. We have replaced the old Figure S2 with these two new plots.

[Figure]

Figure S2: Location of UOG facilities in the vicinity of the Fort McMurray AOSR area from (a) the 2011-based projected 2013 inventory (red dots) superposed on the 2000-based projected 2010 inventory (cyan dots) and (b) the 2000-based projected 2010 inventory (cyan dots) superposed on the 2011-based projected 2013 inventory (red dots). In panel (a) the cyan dots not covered by the red dots are facilities that were in the projected 2010 inventory, but not in the projected 2013 inventory, whereas in panel (b) the red dots not covered by the

cyan dots are facilities that were in the projected 2013 inventory, but not in the projected 2010 inventory. The location of Fort McMurray is marked by the purple diamond symbol

11) Page 12 (Line 20) SMOKE reference has been mentioned – no need to repeat here.

SMOKE reference removed as suggested.

12) Page 13 (Line 14) I agree that the temporal profiles for fugitive VOC are dependent of temperature, but that's not the only factor. Wind speed (and snow cover) can also affect emission flux. How did you address wind speed effects?

This is a very good point. We did not take wind speed (and snow cover) into account for the current study. However we did point out in the Summary and Future Work section that this is an area for future improvement. The text reads:
"In future, model-predicted or locally measured hourly temperature and wind speed may be used to estimate hourly fugitive VOC emissions if the dependence of fugitive VOC emission rates on temperature and wind speed can be parameterized (Li et al., 2017)" (now Page 25, Lines 6-9).
We have now added one more sentence to the paper:
"Snow cover over the mining areas and ice cover over the ponds during wintertime also affect fugitive VOC emissions and need to be considered." (Page 25, Lines 9-10).

13) Page 14 (Line 29 and also Page 15 Line 5) Need a reference to those tests that show 'significant (as the authors claimed)' effects of the artificial water bodies. Be specific about what effects are.

Thank you for the comment. We have added one plot (Figure S3) to show that the change of land use can affect the predicted planetary boundary layer height by a few hundred meters, and we added the following text to the end of the paragraph:
"Meteorological fields are also affected. For example, Figure S3 shows that the predicted planetary boundary layer height over the OS facilities can be a few hundred meters lower than the surrounding areas, similar to the effect of natural lakes." (Page 12, Lines 7-9)

14) Page 15 (2nd paragraph) The authors' effort in updating landuse/land cover is to be commended. This is rarely done in air quality applications. Since BELD4 is already available, would future work require this step?

Thank you very much for recognizing our efforts in this regard. BELD4 is currently available with updated data for the U.S., but MODIS land-use data are still used in BELD4 for Canada with much less detailed land-cover data. We are currently updating the Canadian biogenic emissions land-use database to be consistent with the U.S. BELD4. Once the update is finished, this step will not be needed, at least for the near future. However, it may still be needed for areas that undergo fast development, such as the AOSR.

15) Page 15 (Line 9) Instead of using qualitative description such as 'significantly impacted', present facts. , e.g. change in biogenic VOCs (%).

Thank you for the comment. We have changed the statement as follows:
"By applying these masks to update vegetation and land-cover data, GEM-MACH-calculated biogenic emissions can be reduced by as much as 100% for the cleared areas related to mining activities." (Page 12, Lines 5-7).
This impact can be seen in Figure 7.

16) Page 16 (Line 25; Figure S5, S6) Why not comparing 'annual' CEMS to the NPRI inventory? As the authors noted, there could be large variation in the CEM measurements. Using two-month worth of data does not give me confidence in the comparison shown in Figure S5 and S6. Alternatively, you can report NPRI emissions that are temporally allocated to the same two months.

This is a very good comment. Detailed CEMS data are not required to be reported to the national inventory. We obtained the two months of CEMS data directly from the Alberta provincial government (their assistance is acknowledged in the Acknowledgements section). NPRI annual emissions are reported annually. Facilities are also required to report monthly emissions profiles. We checked the NPRI monthly profiles for $SO_2$, VOC, and $NO_x$ for all six of OS mining facilities for years 2013 to 2016. All but one facility reported constant monthly profiles for all three species for all four years, which is very surprising in light of the major wildfire in May 2016 that ravaged the AOSR area and caused mining operations to be shut down for most of May and June. The monthly profiles reported by the one facility that did not report constant monthly profiles are shown in the table below. We can see that the monthly profiles for this facility also seem questionable. For example, $SO_2$ monthly profiles did not change at all for the first three years. Although the $SO_2$ monthly profile was changed for the fourth year, it only varies by season. VOC profiles are constant for the first three years, but the fourth year has the same seasonal variation as the $SO_2$ and $NO_x$ emissions.

Alternatively it is possible to allocate the NPRI annual emissions to monthly emissions using the production-based derived monthly profiles. However, this approach will not account for $SO_2$ and $NO_x$ emissions during upset events, when large amount of emissions may be released, as indicated from the CNRL daily reports during one upset event. Since these two plots are mainly for "sanity-check" purposes, we decided to annualize the CEMS data with an understanding that there is large uncertainty associated with the comparison by showing the CEMS standard deviation as error bars.

| | SO2 | | | | | VOC | | | | | NOX | | | |
|---|---|---|---|---|---|---|---|---|---|---|---|---|---|---|
| | 2013 | 2014 | 2015 | 2016 | | 2013 | 2014 | 2015 | 2016 | | 2013 | 2014 | 2015 | 2016 |
| **January** | 10.1% | 10.1% | 10.1% | 10.3% | | 8.33% | 8.33% | 8.33% | 10.30% | | 11.40% | 11.40% | 9.20% | 10.30% |
| **February** | 10.1% | 10.1% | 10.1% | 10.3% | | 8.33% | 8.33% | 8.33% | 10.30% | | 8.90% | 8.90% | 9.60% | 10.30% |
| **March** | 7.5% | 7.5% | 7.5% | 10.3% | | 8.34% | 8.34% | 8.34% | 10.30% | | 9.70% | 9.70% | 9.20% | 10.30% |
| **April** | 3.6% | 3.6% | 3.6% | 2.7% | | 8.33% | 8.33% | 8.33% | 2.70% | | 7.40% | 7.40% | 9.80% | 2.70% |
| **May** | 2.6% | 2.6% | 2.6% | 2.6% | | 8.33% | 8.33% | 8.33% | 2.70% | | 3.20% | 3.20% | 8.20% | 2.70% |

| | | | | | | | | | | | | |
|---|---|---|---|---|---|---|---|---|---|---|---|---|
| **June** | 7.6% | 7.6% | 7.6% | 2.7% | | 8.34% | 8.34% | 8.34% | 2.70% | | 8.30% | 8.30% | 7.60% | 2.70% |
| **July** | 7.1% | 7.1% | 7.1% | 9.7% | | 8.33% | 8.33% | 8.33% | 9.60% | | 7.50% | 7.50% | 8.20% | 9.60% |
| **August** | 9.3% | 9.3% | 9.3% | 9.7% | | 8.33% | 8.33% | 8.33% | 9.60% | | 6.70% | 6.70% | 6.50% | 9.60% |
| **September** | 13.5% | 13.5% | 13.5% | 9.7% | | 8.34% | 8.34% | 8.34% | 9.70% | | 7.60% | 7.60% | 7% | 9.70% |
| **October** | 9.7% | 9.7% | 9.7% | 10.6% | | 8.33% | 8.33% | 8.33% | 10.70% | | 8.30% | 8.30% | 8.20% | 10.70% |
| **November** | 8.8% | 8.8% | 8.8% | 10.7% | | 8.33% | 8.33% | 8.33% | 10.70% | | 10.40% | 10.40% | 7.90% | 10.70% |
| **December** | 10.1% | 10.1% | 10.1% | 10.7% | | 8.34% | 8.34% | 8.34% | 10.70% | | 10.60% | 10.60% | 8.60% | 10.70% |

.

17) Page 17 (Line 14) How are the aircraft-estimated VOC emissions spatially allocated within each facility for modeling? The numbers shown in Table 6 are in tonnes/year, but aircraft observations are only available during the field campaign. Hourly aircraft data was scaled to daily rate based on temperature profile during the flight day (Li et al). Were the aircraft-derived estimated annualized by scaling daily values with temperature (as was done in the base case)?

In order to obtain gridded model emissions, the aircraft-estimated facility-total VOC emissions were first split by process based on the process-specific VOC emissions compiled for the base case.   The process-specific emissions for each facility were then spatially allocated within each facility based on the facility-specific and process-specific gridded spatial surrogate fields.

To respond to your question about the annualization calculation for the aircraft-measurement-based VOC estimates, we have added the following sentence to Page 15, Lines 6-7 (note that Table 6 is now Table 4):
"The aircraft-derived VOC emissions estimates shown in Table 4 were annualized by scaling daily values with seasonal variation factors as discussed in Li et al. (2017)."

18) Page 18 (Line 1) The observation-based VOC profiles show lower higher-aromatic fractions. Could this be a result of flight altitude (150-1470 m) which could be more influenced by stack emissions rather than surface sources like tailing ponds?

Total stack VOC emissions are estimated to be very low compared to fugitive emissions from tailings pond, mine faces, and plants.  Therefore, we think it unlikely that the composition of VOC emissions from stacks had a material impact on the observation-based VOC speciation profiles.

19) Page 18 (Line 11) My understanding is that the aircraft-derived emissions are at facility level so any scaling would be the same across all grid cells within individual facilities. Why are the ratios shown in Figure S7 varying within individual facilities?

This is a very good question.  It is due to the different emission rates and different VOC speciation profiles for plants, mine faces, and tailings ponds.  Aircraft-derived VOC profiles

are for the whole facility, but the VOC speciation profiles used for the base case are process-specific, i.e., profiles are different between plants, mine faces, and tailings ponds. Therefore, the ratio of aircraft-observation-based ADOM-2 higher-alkane emissions to the base-case higher-alkane emissions for the GEM-MACH 2.5-km grid varies slightly within the facility due to spatial variations in the base-case emissions field associated with the geographic locations of different processes. We had explained this in the paper on Page 18, Line 11 (now Page 16, Lines 4-5) as "The variations seen within individual facilities are due to different emission rates for plants, mine faces, and tailings ponds". However, in formulating our response here we noticed that the phrase "and different VOC speciation profiles" was missing from that sentence. We have now added it.

20) Page 19 (Line 29) I understand that the aircraft-derived emissions are only used for summertime modeling. The emission comparison needs to be limited to just summer months (so summing basecase emissions for these months), instead of assuming constant daily emissions throughout the year to get annualized value. Reporting 61,500 tpy of PM2.5 is totally misleading and overemphasizing the role of wind-blown dust.

This is a very good point. We agree that the figure of 61,500 t/year is misleading, although we had stated near the beginning of the paragraph that "Note that the latter were annualized for this comparison simply by assuming constant daily emissions throughout the year, which does not account for modulation by snow cover, frozen ground, or precipitation, but the aircraft-observation-based estimates were only used in GEM-MACH for summertime modelling."

Instead of annualizing the aircraft-derived emissions, we have now estimated two-month emissions during the field study period (August and September) based on a similar simple assumption that daily emissions were constant throughout these two warm-season months, which is a more reasonable assumption than assuming constant daily emissions throughout the year. We found that even the estimated aircraft-derived $PM_{2.5}$ emissions for two months are larger than the annual $PM_{2.5}$ emissions from the base-case inventory for all of the facilities except for the Imperial Oil Kearl facility. We think comparing the estimated two-month aircraft-derived emissions with the annual base-case emissions serves the same purpose of showing that the observation-based $PM_{2.5}$ emissions can be much larger than the base-case inventory $PM_{2.5}$ emissions while avoiding the larger uncertainty associated with the annualization calculation. Therefore, we have modified Figure 9. We now compare facility-level $PM_{2.5}$ emissions between the annual, base-case, inventory-based values and the aircraft-observation-based estimates for two summer months for the six AOSR facilities. The wording in the last paragraph of Page 19 (now the second paragraph of Page 17) was also revised accordingly.

21) Page 20 (Line 29) It is interesting to see a high fraction of OC from soil in this area. Is there any literature that reports similar finding?

As far as we can determine, Wang et al. (2015) is the only published report studying the chemical composition of dust over the AOSR area. However, there are a number of "Unpaved Road" dust speciation profiles archived in the U.S. EPA SPECIATE4.5 database that are based on measurements from various locations (see https://www.epa.gov/air-emissions-modeling/speciate-version-45-through-40). The Organic Carbon percentage varies from less than 1% at a California site to more than 20% at a Mexico City site.

22) Summary Section – Information on Phase I and Phase II is repetitive (page 23 and 24). The authors should focus on Phase III and some new insights that this work has revealed. Some recommendations made in this paper are reasonable. Since the basecase and sensitivity emissions were already applied in GEM-MACH, what are the lessons learned? On page 27 (last paragraph), the authors suggest that fugitive dust emissions are underestimated based on air quality modeling (by how much?). The authors should distill some scientific findings to share with readers for other pollutants (such as NOx, VOC, or SO2) based on other air quality modeling activities referred in the paper.

Thank you for this "arms-length" perspective. As suggested, we have shortened the Phase 1 and Phase 2-related discussion by two-thirds. Now the main focus of the paper is on Phase 3 and the lessons learned from this study.

On page 27 (last paragraph, now Page 25, second paragraph), we referred to some GEM-MACH modelling results (Makar et al., 2018) that suggested the fugitive dust emissions, or the fraction of their mass that is composed of base cations, might be underestimated by even more than suggested by the aircraft measurements. However, we were not certain whether this larger underestimate is in fact true and we stated later in the same paragraph that "Further aircraft-based measurements of fugitive dust emissions and their speciation are needed to improve the emissions inventories used here" (now Page 25, Lines 20-21). Nevertheless, we agree that the original phrase "are underestimated" is too strong and therefore we changed it to "might be underestimated" (Page 25, Line 19).

We also agree with the suggestion to summarize scientific findings based on other air quality modelling activities using emissions discussed in this paper. We have revised the paper accordingly. Specifically, we have expanded/added the following statements:

"Akingunola et al. (2018) showed that model-predicted $SO_2$ concentration could be changed by as much as 50% and the $NO_x$ concentration by about 10% using the CEMS-measured hourly stack flow rate and temperature. On the other hand, the use of the more realistic CEMS-measured volume flow rates and temperatures resulted in a slight degradation of model performance with a new, improved plume-rise algorithm." (Page 14, Line 3-7)

"Similar to Ahmadov et al. (2015), Stroud et al. (2018) demonstrated that the measurement-derived top-down emissions improved the modelled VOC and organic aerosol (OA) concentration maxima in plumes. Bias was also improved for OA predictions. Their study

suggested that intermediate volatile organic compound (IVOC) emissions needs to be included as precursors to SOA for further improvement of SOA predictions." (Page 19, Lines 15-18)

"In their examination of acidifying deposition in the region, Makar et al. (2018) found that the new aircraft-based emissions improved the model fit to observations, increasing correlation coefficients (R from 0.47 to 0.54) and improving slopes of the model-to-observation best-fit line (slope changed from 0.051 to 0.73, correcting most of the large underestimate in predicted base cation deposition). The revised fugitive dust estimates from the aircraft study, while resulting in greatly improved model performance relative to the reported emissions, still resulted in an underestimate of base cations relative to observations, implying the need for further improvements to these emissions data." (Page 19, Lines 18-25)

23) Figure 1. This figure needs improvement on print quality/resolution. Table S3. Why was the 2010 CEMA off-road inventory selected? Did you compare these emissions with other inventories? Figure S3. Resolution issue. Color scales are not readable. And why is ethane shown instead of total VOC?

We have replotted Figure 1 with better quality and resolution.

Concerning Table S3 (now Table S4), there were two main reasons for using the 2010 CEMA off-road inventory for Phase 3. First, based the inventory analysis conducted during Phase 1 and Phase 2, we deemed the 2010 CEMA off-road inventory to be the best available inventory for the study area. Second, for Phase 3, the 2013 Canadian APEI emissions were available for all sectors, except for on-road and off-road emissions. Therefore, we had to continue to use the 2010 CEMA off-road emissions. We documented this fact on Page 6, Lines 2-4, as follows:
"2013 Canadian APEI Version 1 from ECCC for all sectors, including the first version of reviewed, publicly-available 2013 NPRI (released December 2014), except for on-road and off-road mobile source emissions (Sassi et al., 2016)."

We also identified the issue with off-road emissions as an area for further improvement in the "Summary and Future Work" section (see the first two paragraphs on Page 24).

Concerning Figure S3 (now Figure S4), we have replotted this figure with better quality and resolution. Ethene (or ethylene, $C_2H_4$) was shown in this figure because it is one of the model speciated VOC species and we used it as a representative for total VOC emissions. However, for the new plot, we have replaced ethene with total VOC as suggested.